# SCALING LAWS FOR IMITATION LEARNING IN SINGLE-AGENT GAMES

## ABSTRACT

Imitation Learning (IL) is one of the most widely used methods in machine learning. Yet, many works find it is often unable to fully recover the underlying expert behavior (Wen et al., 2020; Jacob et al., 2022), even in constrained environments like single-agent games (De Haan et al., 2019; Hambro et al., 2022b). However, none of these works deeply investigate the role of scaling up the model and data size. Inspired by recent work in Natural Language Processing (NLP) (Kaplan et al., 2020; Hoffmann et al., 2022) where "scaling up" has resulted in increasingly more capable LLMs, we investigate whether carefully scaling up model and data size can bring similar improvements in the imitation learning setting for single-agent games. We first demonstrate our findings on a variety of Atari games, and thereafter focus on the extremely challenging game of NetHack. In all games, we find that IL *loss* and *mean return* scale smoothly with the compute budget (FLOPs) and are strongly correlated, resulting in power laws for training compute-optimal IL agents. Finally, we forecast and train several NetHack agents with IL and find they outperform prior state-of-the-art by 2x in all settings. Our work both demonstrates the scaling behavior of imitation learning in a variety of single-agent games, as well as the viability of scaling up current approaches for increasingly capable agents in NetHack, a game that remains elusively hard for current AI systems.

## 1 INTRODUCTION

While conceptually simple, imitation learning has powered some of the most impressive feats of AI in recent years. AlphaGo (Silver et al., 2016) used imitation on human Go games to bootstrap its Reinforcement Learning (RL) policy. Cicero, an agent that can play the challenging game of Diplomacy, used an IL-based policy as an anchor to guide planning (Jacob et al., 2022). Go-Explore, a method for hard-exploration problems which solved all previously unsolved Atari games, used self-imitation learning in its robustification phase (Ecoffet et al., 2021).

Despite its prevalence, several works have pointed out some of the limitations of IL. De Haan et al. (2019) and Wen et al. (2020) call out the issue of *causal confusion*, where the IL policy relies on spurious correlations to achieve high training and held-out accuracy, but performs far worse than the data-generating policy, even in single-agent Atari games. Jacob et al. (2022) have mentioned similar issues for policies learning from human games: *they consistently underperform the data-generating policy*. However, in many of these works, the role of model and data size is not deeply investigated. This is especially striking considering the increasingly impressive capabilities that recent language models have exhibited, mostly as a consequence of scale. In a series of papers trying to characterize these improvements with scale starting with Hestness et al. (2017) and Rosenfeld et al. (2019), it has been shown language modeling loss (i.e. cross-entropy) scales smoothly with model size and number of training tokens (Kaplan et al., 2020; Hoffmann et al., 2022). If we think of language models as essentially performing "imitation learning" on text, then a natural next question is whether some of these results extend to IL-based agents in games, and whether scale could provide similar benefits and alleviate some of the issues mentioned earlier on.

In this paper, we ask the following question: *How does compute in terms of model and data size affect the performance of agents trained with imitation learning in the single-agent game setting?* We first focus on several Atari games with dense rewards which allows us to demonstrate our findings on a variety of games. However, since Atari games have all been solved at this point, there is not

much room for further improvement in terms of scaling up. To demonstrate the potential of scaling up IL, our core focus will be on the extremely challenging game of NetHack, a roguelike video game released in 1987. NetHack is an especially well-suited and interesting domain to study for several reasons. First, it is procedurally generated and highly stochastic, disqualifying approaches relying heavily on memorization instead of generalization, such as Go-Explore (Ecoffet et al., 2021). Second, the game is partially observed, requiring the use of memory, potentially for thousands of steps due to the game's long-term dependencies. Finally, the game is extremely challenging for current AI systems, with current agents reaching scores nowhere close to average human performance[1]. The best agent on NetHack is a purely *rule-based* system called AutoAscend (Hambro et al., 2022a), with RL approaches lagging behind (Hambro et al., 2022b; Küttler et al., 2020; Mu et al., 2022; Mazoure et al., 2023). Even just recovering this system is hard, with Hambro et al. (2022b) reporting that the best neural agents achieve less than 10% of the system's mean return in the environment, causing the authors to call for significant research advances. We instead investigate whether simply scaling up BC can help close some of this gap.

**Contributions.** We train a suite of neural Atari and NetHack agents with different model sizes using BC to imitate expert policies and analyze the loss and mean return isoFLOP profiles. We find the optimal cross-entropy loss scales as a power law in the compute budget, and we use two different methods to derive scaling laws for the loss-optimal model and data sizes. We then relate the cross-entropy loss of our trained BC agents to their respective mean return when rolled out in the environment, and find that the mean return follows a power law with respect to the optimal cross-entropy loss, showing improvements in loss predictably translate in better performing agents. We use our two scaling law derivations to forecast the training requirements of a compute-optimal neural BC agent for NetHack. These forecasts are then used to train an agent which outperforms prior neural NetHack agents by 2x in all settings, showing scale can provide dramatic improvements in performance. We briefly extend our results to the RL setting, where we also train a suite of NetHack agents using IMPALA (Espeholt et al., 2018) and again find that model and data size scale as power laws in the compute budget.

Our results demonstrate that the improvements in imitation learning performance for single-agent games with dense rewards[2] can be described by clean power laws. This suggests carefully scaling up model and data size can provide a promising path towards increasingly capable game agents, as well as potentially boost performance in other imitation learning settings.

## 2 PRELIMINARIES

We now introduce the formal setup for behavioral cloning. We assume the environment can be described by a Partially Observable Markov Decision Process (POMDP) $\langle S, T, A, O, R, \gamma \rangle$, with states $S$, transition function $T$, action set $A$, possible observation emissions $O$, reward function $R(s, a)$, and discount factor $\gamma$.

In the behavioral cloning setup, we don't assume access to the rewards but instead assume access to a dataset $\mathcal{D}$ consisting of trajectories $\tau = (s_0, a_0, s_1, a_1, \ldots)$ of states and actions. These trajectories can be generated by multiple (possibly sub-optimal) demonstrators acting in the environment. However, in this work, they are assumed to all come from the same expert policy $\pi$. The goal is to recover this expert policy. To do this, a learner $\pi_\theta$ will optimize the following cross-entropy loss:

$$\mathcal{L}(\theta) = -\mathbb{E}_{(h_t, a_t) \sim \mathcal{D}} \left[ \log \pi_\theta(a_t | h_t) \right], \tag{1}$$

where $h_t$ can include part or the entirety of the history of past states and actions.

## 3 EXPERIMENTAL SETUP

We analyzed the scaling behavior of agents trained with BC in two domains: (1) Atari and (2) NetHack. The former serves to test the validity of the scaling laws in a range of games, while the latter tests the performance gains of scaling in an extremely challenging and unsolved game.

---

[1]The average overall human performance is around 127k (Hambro et al., 2022b), while the current best performing NetHack agent gets a score of 10k.

[2]Please refer to section 6 for a discussion of why we need this requirement.

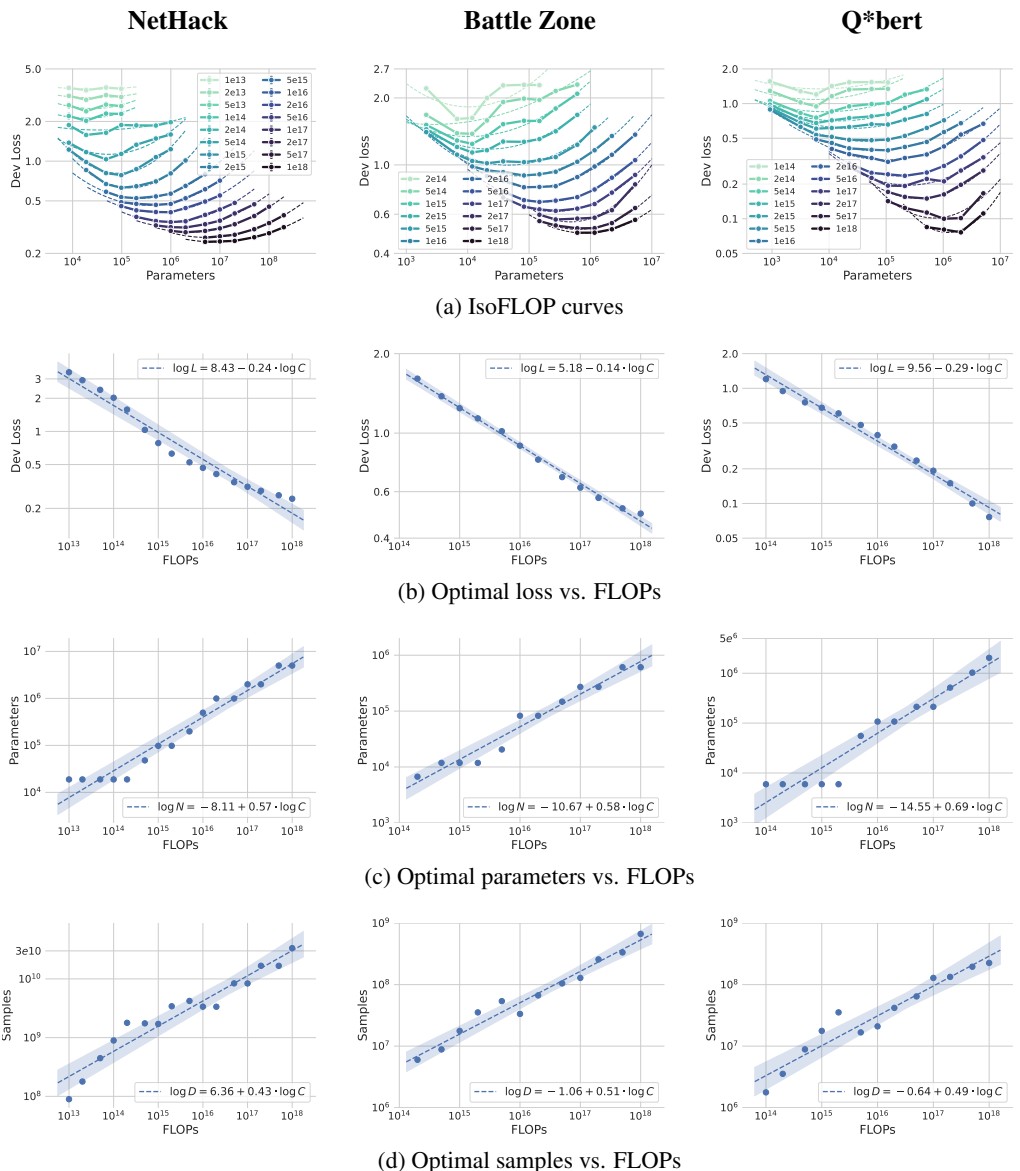

Figure 1: **BC loss scaling.** We train a wide range of model sizes across several orders of magnitudes of FLOP budgets. We plot the validation loss for each model, with fitted parabolas per IsoFLOPs curve (**a**). We then regress the loss minima (**b**), the loss-optimal number of parameters (**c**), and the loss-optimal number of samples (**d**) on their corresponding FLOP budgets. We find clear power law trends for Nethack (first column), Battle Zone (middle column), and Q*bert (last column). The full list of Atari results can be found in Appendix I.

Whenever we report FLOP or parameter counts, we are referring to their *effective* counts, which we define as only including the parts of the network that are being scaled, similar to Hilton et al. (2023) (see Appendix E for full details). Please see Appendix F for details on all hyperparameters.

## 3.1 ATARI

We chose the following set of 8 Atari games: Battle Zone, Q*bert, Bank Heist, Boxing, Breakout, Name This Game, Phoenix, and Space Invaders. We chose these games either because they were

part of the Atari-5 subset[3] (Aitchison et al., 2023), a reduced dataset that aims to be representative of the full set, or they were games where the reward is at least somewhat dense (see section 6 for more discussion on this). We then perform the following steps for each game. First, we train a CNN-based agent with PPO (Schulman et al., 2017) in order to get an expert agent. Second, we gather a dataset of about 1B samples consisting of rollouts of the expert agent. We then train a family of CNN-based agents on this dataset using BC, varying the width of the core CNN and the final linear layer (see Appendix E). The total number of parameters ranged from 1k to 5M.

## 3.2 NETHACK

We train LSTM-based agents on the NLD-AA dataset (Hambro et al., 2022b), mainly varying the width of the LSTM (see Appendix E). The total number of parameters ranged from 10k to 500M. While the original NLD-AA dataset already contains around 3B samples, we extended the dataset to around 60B samples (NLD-AA-L) and 150B samples (NLD-AA-XL) by collecting more rollouts from AutoAscend (i.e. the data-generating policy). NLD-AA-L is used for the results in Figure 1a, while NLD-AA-XL is used for all our forecasting-based experiments (see section 5).

## 4 SCALING UP IMITATION LEARNING

This section is structured as follows. We first investigate the role of model size and number of samples with respect to cross-entropy loss (subsection 4.1). While intuitively it feels like a lower loss should result in a better agent, we verify this by directly investigating the role of model size and number of samples with respect to the environment return (subsection 4.2), and relating these results to the loss results. Finally, we also show a possible extension of our analysis to the RL setting (subsection 4.3).

## 4.1 SCALING LAWS FOR BC LOSS

To investigate the role of model size and number of samples with respect to cross-entropy loss, we follow similar approaches to the ones used in Hoffmann et al. (2022).

**Approach #1: isoFLOP profiles.** "IsoFLOP" refers to constant FLOP budget contour lines. For Atari, we train up to 12 different model sizes, ranging from 1k to 5M. For NetHack, we train 14 different model sizes, ranging from 10k to 500M. For all domains, we train FLOP budgets of at least $1e13$ and up to $1e18$. In Figure 1 we plot the loss evaluated on a held-out set of about 100 trajectories against the parameter count for each FLOP budget. Similarly to Hoffmann et al. (2022), we observe clear parabolas with well-defined minima at the optimal model size for a given compute budget in all games. We take these loss-optimal data points to fit three regressions: one that regresses the log parameters on the log FLOPs, another that regresses the log samples on the log FLOPs, and a final one that regresses the log loss on the log FLOPs. These regressions give rise to the following power laws (Figure 1c, Figure 1d, and Figure 1b):

$$N_{\text{opt}} \propto C^\alpha \quad D_{\text{opt}} \propto C^\beta \quad L_{\text{opt}} \propto C^\gamma, \tag{2}$$

where $N_{\text{opt}}$ indicates the loss-optimal model size, $D_{\text{opt}}$ the loss-optimal number of training samples, $L_{\text{opt}}$ the minimal validation loss, and $C$ the compute budget in FLOPs. We refer to the legends of Figure 1c, Figure 1d, and Figure 1b for sample values of $\alpha$, $\beta$, and $\gamma$, respectively.

**Approach #2: parametric fit.** Instead of only fitting the loss-optimal points as was done in approach #1 above, one can also fit all points from Figure 1a to the following quadratic form:

$$\log \hat{L}(N, D) = \beta_0 + \beta_N \log N + \beta_D \log D + \beta_{N^2} (\log N)^2 + \beta_{ND} \log N \log D + \beta_{D^2} (\log D)^2. \tag{3}$$

If we only look at the linear terms here, we notice that this loss has the form of a Cobb-Douglas production function:

$$\hat{L}(N, D) = \exp(\beta_0) \times N^{\beta_N} \times D^{\beta_D}, \tag{4}$$

---

[3] We also experimented with Double Dunk, which is part of the Atari-5, but found even our smallest models could perfectly learn the policy with very few samples for the expert we trained. Therefore, we left it out.

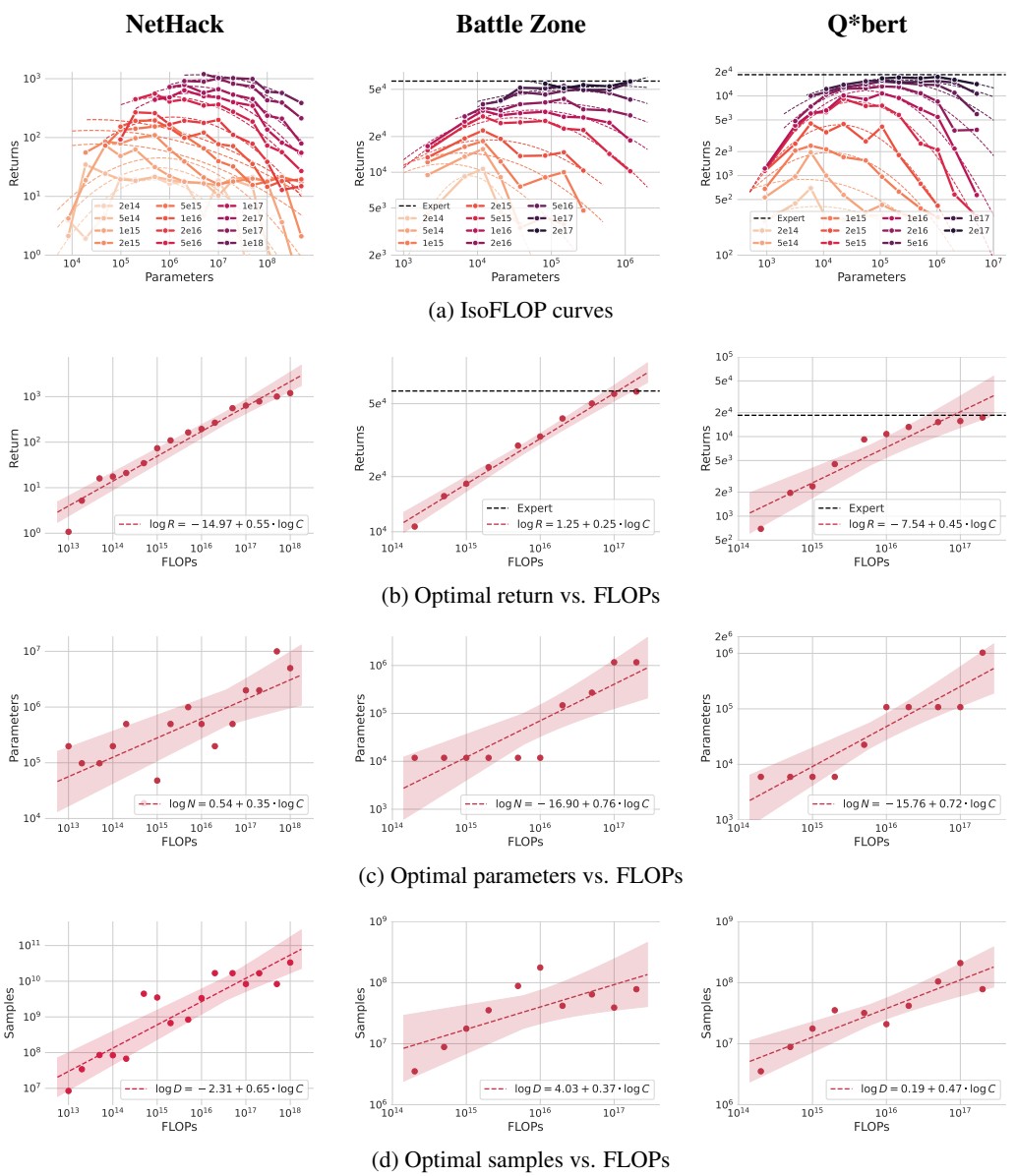

(a) IsoFLOP curves

(b) Optimal return vs. FLOPs

(c) Optimal parameters vs. FLOPs

(d) Optimal samples vs. FLOPs

Figure 2: **BC return scaling.** We train a wide range of model sizes across several orders of magnitudes of FLOP budgets (same models as in Figure 1a) and plot their average return in the environment (**a**). We left off the first four FLOP budgets as we found them to be especially noisy. We then regress the optimal returns (**b**), the return-optimal number of parameters (**c**), and the return-optimal number of samples (**d**) on their corresponding FLOP budgets. We find mostly clear power law trends for Nethack (left), Battle Zone (middle), and Q*bert (right). Full Atari results can be found in Appendix I.

where we can think of parameters $N$ and samples $D$ as inputs that affect how much output (i.e. loss) gets produced. We then take the functional form in Equation 3 and minimize the loss subject to the constraint that $\text{FLOPs}(N, D) \approx 6ND^4$. To do this, we used the method of Lagrange multipliers to get the following functional forms for $N_{\text{opt}}$ and $D_{\text{opt}}$ (see Appendix A for full derivation):

---

[4]Note that this FLOPs equation is only valid for our NetHack experiments, since the model there is LSTM-based. To carry out a similar analysis for Atari, where the models are CNN-based, this formula needs to be adjusted. We only perform the analysis for NetHack due to the simplicity of the FLOPs equation.

Table 1: **Fitted power law coefficients in NetHack.** We list the scaling coefficients for model size ($\alpha$) and number of samples ($\beta$) for all three settings. 95% CIs are noted in parentheses, where the delta method was used for the parametric fit parameters (see Appendix G).

| Setting | IsoFLOP profiles | | Parametric fit | |
|---|---|---|---|---|
| | $\alpha$ | $\beta$ | $\alpha$ | $\beta$ |
| 1. BC Loss | 0.57 (0.50, 0.64) | 0.43 (0.36, 0.50) | 0.48 (0.47, 0.49) | 0.52 (0.51, 0.53) |
| 2. BC Return | 0.35 (0.18, 0.52) | 0.65 (0.48, 0.82) | 0.34 (0.33, 0.35) | 0.66 (0.65, 0.67) |

$$N_{\text{opt}} = G \left( \frac{C}{6} \right)^{\alpha}, \quad D_{\text{opt}} = G^{-1} \left( \frac{C}{6} \right)^{\beta}, \quad \text{where} \quad G = \exp \left( \frac{\beta_D - \beta_N}{2\beta_{D^2} - 2\beta_{ND} + 2\beta_{N^2}} \right). \tag{5}$$

We find that $\alpha = \frac{2\beta_{D^2} - \beta_{ND}}{2\beta_{D^2} - 2\beta_{ND} + 2\beta_{N^2}}$ and $\beta = \frac{2\beta_{N^2} - \beta_{ND}}{2\beta_{D^2} - 2\beta_{ND} + 2\beta_{N^2}}$. We compare the two approaches for NetHack in Table 1.

### 4.2 SCALING LAWS FOR BC RETURN

Note that the analysis in the previous section was all in terms of cross-entropy loss. However, in the imitation learning setting, we almost never care directly about this quantity. Instead, we care about the average return of the resulting agent in the environment. To investigate how this quantity scales, we roll out every model from Figure 1a in the corresponding Atari or NetHack environment and average their score across 100 (Atari) and 1k (NetHack) rollouts each. We show the results in Figure 2a. We then follow a similar procedure as in subsection 4.1 and perform the same three regressions, giving rise to the following power laws (Figure 2c, Figure 2d, and Figure 2b):

$$N_{\text{opt}} \propto C^{\alpha} \quad D_{\text{opt}} \propto C^{\beta} \quad R_{\text{opt}} \propto C^{\gamma}, \tag{6}$$

where $N_{\text{opt}}$ indicates the return-optimal model size, $D_{\text{opt}}$ the return-optimal data size, $R_{\text{opt}}$ the maximal return, and $C$ the compute budget in FLOPs. We refer to the legends of Figure 2c, Figure 2d, and Figure 2b for sample values of $\alpha$, $\beta$, and $\gamma$, respectively. When looking at Figure 2b, we find that for the Atari games the power laws hold all the way until expert performance[5]. For NetHack, we find more FLOPs will be required to reach the expert score of 10k.

Additionally, we can take the functional form in Equation 3 and simply replace loss with mean return. We can then solve the same constrained optimization problem resulting in the exact same expressions as found in Equation 5. We list the resulting coefficients for NetHack in Table 1.

To investigate the relationship between loss and mean return, we regress the loss-optimal log returns on the corresponding log loss values. We find a power law of the form $R_{\text{opt}} \propto L_{\text{opt}}^{\delta}$, as shown in Figure 3. The fit in the figure shows optimal loss and mean return are highly correlated in all games, indicating we can expect return to increase smoothly as we make improvements in loss, rather than showing sudden jumps.

### 4.3 EXTENSION TO REINFORCEMENT LEARNING

Given the stark trends we found for BC in the previous sections, we investigate whether similar trends can be found for RL. We explore this briefly for the game of NetHack since several works in the past years have attempted RL-based approaches for NetHack (Küttler et al., 2020; Hambro et al., 2022b) without too much success, unlike in Atari. We investigate the role of model size and environment interactions using approaches 1 and 2 from subsection 4.1 applied to IMPALA (Espeholt et al., 2018).

While learning curves in RL tend to have high variance, Figure 4 suggests that compute-optimal agents should increase both the number of parameters and number of environment interactions as the

---

[5]Note that the breaking down of our scaling laws *after* reaching expert performance is *expected*. This is similar to other scaling laws such as those of Kaplan et al. (2020) breaking down at the entropy of language.

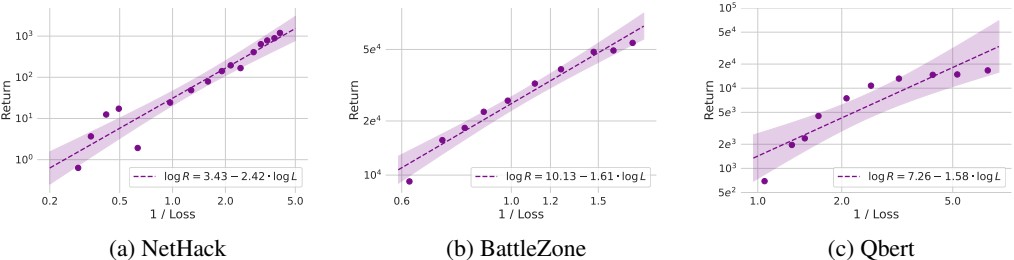

Figure 3: **BC return vs. optimal loss.** We investigate the relationship between the optimal loss of a BC agent and the mean return. We find they are highly correlated for all games.

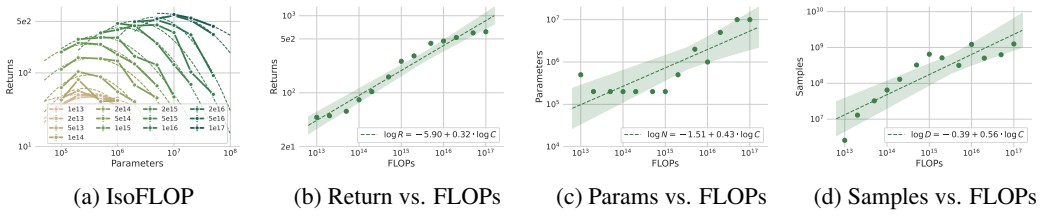

Figure 4: **RL return scaling.** We train a wide range of model sizes across several orders of magnitude of FLOP budgets and plot the average return when rolled out in the environment at the end of training (**a**). We then regress the return-optimal average returns (**b**), parameters (**c**), and samples (**d**) on their corresponding FLOP budgets. We run 1 seed per point on the isoFLOP profile.

FLOP budgets are scaled up. We also find that the NetHack game score varies smoothly with FLOPs and hence can be seen as a *natural performance metric* (Hilton et al., 2023). We provide complete details of our setup and results in Appendix H.

## 5 FORECASTING COMPUTE-OPTIMAL BC AGENTS

The isoFLOP profiles and power laws shown in Figure 1 and Figure 2 allow us to estimate the compute-optimal number of samples and parameters needed to train an agent that recovers the expert's behavior. For all our Atari games except for Space Invaders, we already found such an agent! It is simply the first dot that reaches the expert score in Figure 2b. However, for NetHack, even the largest FLOP budget models don't come close to expert performance (10k). To attempt to close this gap, we forecast and train a compute-optimal NetHack agent aimed at getting a score of 10k.

To do this, we follow two approaches:

1. **Using loss isoFLOP profiles.** We first plug in $R = 10k$ into the regression in Figure 3a to solve for $L_{10k}$, the loss needed for a score of $10k$. Then, we plug $L_{10k}$ into the regression in Figure 1b to get $C_{10k}$, the FLOPs needed to recover a score of $10k$. Then we find the optimal parameters and samples using Figure 1c and Figure 1d. This way, we find that the model size should be 43M, and the data size should be 144B.

2. **Using parametric fit.** We take $C_{10k}$ from above, and use Equation 5 found by the parametric fit to solve for the parameters $N$ and samples $D$. This way, we find that the model size should be 17M, and the data size should be 362B.

We find that the second approach predicts a smaller model size but more samples, similar to findings in Hoffmann et al. (2022). Based on early forecasting fits, we train a 30M parameter model for 115B samples, which took 11 days on 8 GPUs. The results can be found in Table 2. While we do not recover the underlying expert behavior (score of 10k), we do find the resulting model gets a big boost in performance and outperforms prior state-of-the-art by 2x, both when using a random initial character (hardest setting) as well as when its kept fixed to human monk.

**Discussion** The gap with the expert could have several explanations. Uncertain power law exponents may have caused substantial extrapolation error when predicting model and data sizes for FLOP budgets much larger than those in the isoFLOP profile. In Appendix J, we perform a rolling cross-validation to evaluate one-step-ahead forecasting performance, which we do find to be accurate.

Table 2: **Forecasting results.** We compare a model trained with BC using 30M parameters on 115B samples with previous models in the `NetHackChallenge-v0` environment and find it outperforms all of them on both randomized character initialization (harder) as well as on human monk (easier). *Exact scores not reported. Scores from Hambro et al. (2022b) were adjusted to account for an error in their evaluation code. See Appendix B for full results with standard errors.

| Models | All Random | Human Monk |
|---|---|---|
| **Offline only** | | |
| DQN-Offline (Hambro et al., 2022b) | 0.0 | 0.0 |
| CQL (Hambro et al., 2022b) | 352 | 366 |
| IQL (Hambro et al., 2022b) | 171 | 267 |
| BC (CDGPT5) (Hambro et al., 2022b;a) | 554 | 1059 |
| BC (Transformer) (Piterbarg et al., 2023) | 1318 | - |
| **Scaled-BC (ours)** | **2740** | **5218** |
| **Offline + Online** | | |
| Kickstarting + BC (Hambro et al., 2022b) | 962 | 2090 |
| APPO + BC (Hambro et al., 2022b) | 1282 | 2809 |
| APPO + BC (Piterbarg et al., 2023) | 1551 | - |
| LDD* (Mu et al., 2022) | - | 2100 |

# 6 LIMITATIONS

**Natural performance metrics.** There is no reason in general to expect game scores to scale smoothly. If they do, Hilton et al. (2023) define them as *natural performance metrics*. We expect that for any game score to be a natural performance metric, it needs to be at least somewhat dense so it tracks learning progress, which is why we focused on environments with relatively dense rewards in this paper[6]. It's possible our results extend to highly sparse reward settings as well, but one may need to introduce alternative proxy metrics (e.g. intrinsic performance (Hilton et al., 2023)) in that case.

**Experimental setup.** Previous works have pointed to the importance of tuning hyperparameters (e.g. learning rate, batch size, adam optimizer parameters, etc.) for every run on the isoFLOP profile. Since we didn't find any major sensitives to hyperparameters during some initial tuning, and to limit computational cost, we kept all hyperparameters fixed for all isoFLOP profiles (Figure 1a, Figure 2a, and Figure 4a) and used "snapshots" of the same run to evaluate different FLOP budgets for the same model size. Therefore, we would like to point out there is considerable uncertainty in the exact values of the reported power law coefficients. Nevertheless, we expect the overall trends to still hold.

# 7 RELATED WORK

**NetHack** Work on NetHack has been quite limited so far, with early work establishing the NLE benchmark (Küttler et al., 2020), evaluating symbolic vs. neural agents (Hambro et al., 2022a), and creating large-scale datasets based off of rule-based and human playthroughs for methods aiming to learn from demonstrations (Hambro et al., 2022b). More recent work has either focused on better reward signal supervision and sample efficiency through proxy metrics and contrastive pre-training (Mazoure et al., 2023; Bruce et al., 2023) or leveraged dynamics models with language descriptions in order to improve sample efficiency and generalization (Mu et al., 2022). Concurrent

---

[6]This, however, does *not* guarantee we will observe scaling laws in these environments when using IL!

work also investigates the gap between neural methods and AutoAscend, but focuses on leveraging an action hierarchy, improvements in architecture, and fine-tuning with RL (Piterbarg et al., 2023).

**Scaling laws**   Hestness et al. (2017) and Rosenfeld et al. (2019) are one of the earliest works that try to characterize empirical scaling laws for deep learning. Kaplan et al. (2020) and Hoffmann et al. (2022) specifically focus on training compute-optimal language models, finding similar trends as presented in this paper. While in the imitation learning setting, our agents also minimize cross-entropy loss, we additionally show that the eventual performance of the agent as measured by the average return in the environment scales smoothly with the loss. Other works focus more broadly on generative modeling (Henighan et al., 2020), or analyze specific use cases such as acoustic modeling (Droppo & Elibol, 2021). Clark et al. (2022) investigate scaling laws for routing networks, and Hernandez et al. (2021) study scaling laws for transfer, finding the *effective data transferred* (the amount of extra data required to match a pre-trained model from scratch) follows a power-law in the low-data regime. More recent works have also tried to extend these scaling law results to multi-modal learning (Cherti et al., 2022; Aghajanyan et al., 2023). Caballero et al. (2022) introduce *broken neural scaling laws*, which allow modeling of double descent and sharp inflection points. Finally, scaling laws relate to sample complexity theory, which shows that increases in the number of samples (i.e. dataset size) can improve the suboptimality of IL (Rajaraman et al., 2020; Xu et al., 2020; Rajaraman et al., 2021) and is applicable to all architectures and datasets.

Perhaps the closest work to our paper is that of Hilton et al. (2023), who characterize scaling laws in RL. However, they don't consider IL, and they do not evaluate on Atari or NetHack, the latter of which we consider an especially interesting environment because of its extremely challenging nature.

## 8   DISCUSSION

**Extensions beyond single-agent games.**   We have shown that in the imitation learning (and to some extend in the reinforcement learning setting), scaling up model and data size provides predictable improvements, and a promising path to improving performance, as demonstrated in a variety of Atari games and in the full game of NetHack. While we do not extend our analysis beyond single-agent games in this paper, we believe these results could be suggestive of similar findings across many imitation learning domains, where oftentimes model and data sizes are not carefully picked.

**Leveraging human data.**   In this work, we did not consider analyzing the scaling relationships when using human trajectories (e.g. from NLD-NAO (Hambro et al., 2022b)) instead of those from AutoAscend (NLD-AA (Hambro et al., 2022b)). This is because extra care must be taken to handle the lack of actions in the human dataset, requiring techniques such as BCO (Torabi et al., 2018). Investigating scaling laws here could be especially interesting since: (1) the human dataset is more diverse, containing trajectories from many different players with varying level of skill, and (2) it contains many examples of trajectories that ascend (i.e. win the game). (1) could shed perspective on the role of scaling when the data includes many different and potentially suboptimal demonstrations, similar to Beliaev et al. (2022). (2) could provide insight into the viability of methods such as Video PreTraining (Baker et al., 2022) since these rely heavily on being able to clone the expert data well.

## 9   CONCLUSION

In this work, we find that imitation learning loss and mean return follow clear power law trends with respect to FLOPs, as demonstrated in Atari and in the challenging game of NetHack. In addition, we find loss and mean return to be highly correlated, meaning improvements in loss predictably translate in improved performance in the environment. Using the found power laws, we forecast the compute requirements (in terms of model and data size) to train compute-optimal agents aimed at recovering the underlying expert. In NetHack, we find the performance improves dramatically, surpassing prior SOTA by 2x in all settings. We also briefly extend our results to the reinforcement learning setting, and find similar power laws for model size and number of interactions in NetHack. Our results demonstrate that scaling up model and data size is a promising path towards training increasingly capable agents for single-agent games. More broadly, they also call for work in the larger imitation learning and reinforcement learning community to more carefully consider and study the role of scaling laws, which could provide large improvements in many other domains.

## 10 ETHICS STATEMENT

While we do not see a direct path towards any negative applications, we note that scaling up could have unknown unintended consequences. As scaling results in imitation and reinforcement learning agents that are increasingly more capable and influential in our lives, it will be important to keep them aligned with human values.

## 11 REPRODUCIBILITY STATEMENT

Due to legal reasons, we unfortunately cannot release the code for the NetHack results. However, we included the code for all Atari results as part of the supplementary material. In addition, we plan to release the pretrained weights of the forecasted NetHack agent ( section 5). Finally, we have dedicated several sections in the appendix to ensure reproducibility of our results. Appendix F provides a complete account of all training details, including hyperparameters, dataset information, GPU types, training times, etc. Appendix D provides complete details of our architectures for both domains. Appendix E provides details on how we scale our networks and how we do FLOP counting.

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

## A  PARAMETRIC FIT: POWER LAW DERIVATION

In this section, we show the derivation resulting in Equation 5. We first restate Equation 3 for convenience:

$$\log \hat{L}(N, D) = \beta_0 + \beta_N \log N + \beta_D \log D + \beta_{N^2} \log^2 N + \beta_{ND} \log N \log D + \beta_{D^2} \log^2 D.$$

Now, to minimize this equation with the constraint that $\text{FLOPs}(N, D) = C \approx 6ND$, we use Lagrange multipliers. We first write down the Lagrangian:

$$\mathcal{L}(N, D, \lambda) = \hat{L}(N, D) - \lambda(g(N, D) - C),$$

where $g(N, D) = 6ND$. Now, setting $\nabla \mathcal{L} = \mathbf{0}$ we get:

$$\nabla \log \hat{L}(N, D) = \lambda \nabla(6ND) \quad \text{and} \quad 6ND = C.$$

The former results in the following system of equations:

$$\frac{\partial \log \hat{L}(N, D)}{\partial N} = \lambda \frac{\partial(6ND)}{\partial N}$$

$$\frac{\partial \log \hat{L}(N, D)}{\partial D} = \lambda \frac{\partial(6ND)}{\partial D}.$$

This means that

$$\beta_N \frac{1}{N} + \beta_{ND} \log D \frac{1}{N} + 2\beta_{N^2} \log N \frac{1}{N} = \lambda 6D$$

$$\beta_D \frac{1}{D} + \beta_{ND} \log N \frac{1}{D} + 2\beta_{D^2} \log D \frac{1}{D} = \lambda 6N.$$

Multiplying the top equation by $N$ and the bottom one by $D$ we have that

$$\beta_N + \beta_{ND} \log D + 2\beta_{N^2} \log N = \beta_D + \beta_{ND} \log N + 2\beta_{D^2} \log D.$$

Recalling $C = 6ND$, we solve for $N$ and $D$ in terms of $C$, giving the results listed in Equation 5.

## B  FULL FORECASTING RESULTS

The full set of forecasting results in NetHack can be found in Table 3.

Table 3: **Forecasting results (full).** Table 2 with added standard errors. Results from Hambro et al. (2022b) use 10 seeds, those from Piterbarg et al. (2023) use 6 seeds, while ours are 1 seed due to computational cost. *Exact scores not reported in original work. Scores from Hambro et al. (2022b) were adjusted to account for an error in their original evaluation code.

| Models | All Random | Human Monk |
|---|---|---|
| **Offline only** | | |
| DQN-Offline (Hambro et al., 2022b) | $0.0 \pm 0.0$ | $0.0 \pm 0.0$ |
| CQL (Hambro et al., 2022b) | $352 \pm 18$ | $366 \pm 35$ |
| IQL (Hambro et al., 2022b) | $171 \pm 6$ | $267 \pm 28$ |
| BC (CDGPT5) (Hambro et al., 2022b;a) | $554 \pm 45$ | $1059 \pm 159$ |
| BC (Transformer) (Piterbarg et al., 2023) | $1318 \pm 38$ | - |
| **Scaled-BC (ours)** | $\mathbf{2740 \pm}$ - | $\mathbf{5218 \pm}$ - |
| **Offline + Online** | | |
| Kickstarting + BC (Hambro et al., 2022b) | $962 \pm 50$ | $2090 \pm 123$ |
| APPO + BC (Hambro et al., 2022b) | $1282 \pm 87$ | $2809 \pm 103$ |
| APPO + BC (Piterbarg et al., 2023) | $1551 \pm 73$ | - |
| LDD* (Mu et al., 2022) | - | $2100 \pm$ - |

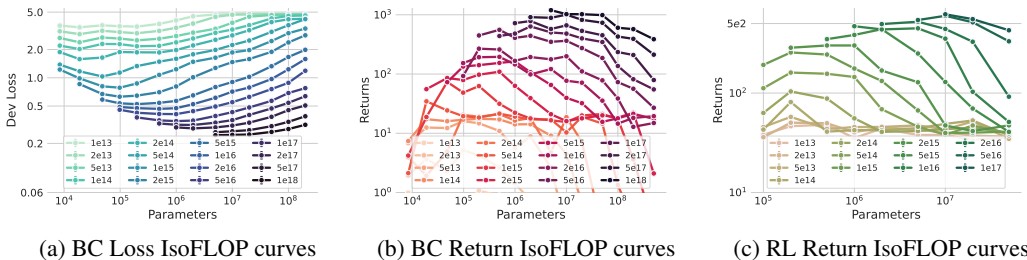

(a) BC Loss IsoFLOP curves      (b) BC Return IsoFLOP curves      (c) RL Return IsoFLOP curves

Figure 5: **Full isoFLOP curves.** (**a**) is similar to Figure 1a but including the full set of points. (**b**) is similar to Figure 2a but including the full set of points. (**c**) is similar to Figure 4a but including the full set of points.

## C  NETHACK FULL SET OF FIGURES

We show the full set of IsoFLOP curves in NetHack for all settings in Figure 5.

## D  ARCHITECTURE DETAILS

### D.1  ATARI

Our architecture for BC experiments in Atari is simple. It consists of the following layers:

1. A convolutional layer with 8 x 8 filter size and stride 4, followed by a ReLU activation layer.
2. A convolutional layer with 4 x 4 filter size and stride 2, followed by a ReLU activation layer.
3. A convolutional layer with 3 x 3 filter size and stride 1, followed by a ReLU activation layer.
4. A final linear layer that maps the flattened output dimension of the CNN layers above to the number of actions in the respective Atari game.

### D.2  NETHACK

We use two main architectures for all our experiments, one for the BC experiments and another for the RL experiments.

**BC architecture.** The NLD-AA dataset (Hambro et al., 2022b) is comprised of *ttyrec*-formatted trajectories, which are $24 \times 80$ ASCII character and color grids (one for each) along with the cursor position. To encode these, we modify the architecture used in Hambro et al. (2022b), resulting in the following:

- **Dungeon encoder.** This component encodes the main observation in the game, which is a $21 \times 80$ grid per time step. Note the top row and bottom two rows are cut off as those are fed into the message and bottom line statistics encoder, respectively. We embed each character and color in an embedding lookup table, after which we concatenate them and put them in their respective positions in the grid. We then feed this embedded grid into a ResNet, which consists of 2 identical modules, each using 1 convolutional layer followed by a max pooling layer and 2 residual blocks (of 2 convolutional layers each), for a total of 10 convolutional layers, closely following the setup in Espeholt et al. (2018).

- **Message encoder.** The message encoder takes the top row of the grid, converts all ASCII characters into a one-hot vector, and concatenates these, resulting in a $80 \times 256 = 20,480$ dimensional vector representing the message. This vector is then fed into a 2-layer MLP, resulting in the message representation.

- **Bottom line statistics.** To encode the bottom line statistics, we flatten the bottom two rows of the grid and create a "character-normalized" (subtract 32 and divide by 96) and "digits-normalized" (subtract 47 and divide by 10, mask out ASCII characters smaller than

45 or larger than 58) input representation, which we then stack, resulting in a $160 \times 2$ dimensional input. This closely follows the Sample Factory[7] model used in Hambro et al. (2022b).

After the components above are encoded, we concatenate all of them together. Additionally, we also concatenate the previous frame's action representation (coming from an embedding lookup table), and a crop representation (a $9 \times 9$ crop around the player, processed by a 5-layer CNN). We then feed this combined representation into a 2-layer MLP, after which a single layer LSTM processes the representation further. Finally, we have two linear heads on top of the LSTM, one for the policy and one for the value (not used for BC).

**RL architecture.** We modify the architecture from Küttler et al. (2020) to also include a 5-layer 1-dimensional CNN that processes the message, as well as another 5-layer 2-dimensional CNN that processes a $9 \times 9$ crop of the dungeon grid around the player.

# E    FLOP AND PARAMETER COUNTING

As mentioned in the main text, we only count FLOPs and parameters for the parts of the model being scaled.

## E.1    ATARI

The model network only consists of three convolutional layers interleaved with ReLUs (see Appendix D for details), followed by a linear layer (i.e. the policy head). We simply scale the width of all layers. This means we scale the channels of the convolutional network, and the width of the linear head on top.

Since we scale the whole network, we count the FLOPs from all convolutional and linear layers. We compute the forward FLOPs of a convolutional layer as $2 * h_{\text{out}} * w_{\text{out}} * c_{\text{out}} * p$, where $h_{\text{out}}$ is the height of the output shape, $w_{\text{out}}$ is the width of the output shape, $c_{\text{out}}$ are the number of output channels, and $p$ indicates the number of parameters in one filter of the current layer (without counting bias). Hence, $p = k^2 * c_{\text{in}}$, where $k$ is the kernel size and $c_{\text{in}}$ is the number of input channels. Following prior work (Hilton et al., 2023), we assume the backward pass takes about twice the number of FLOPs from the forward pass.

## E.2    NETHACK

For all our BC and RL experiments, we only scale the following parts of the model and keep the rest fixed:

- The hidden size of the two-layer MLP right before the LSTM.

- The hidden size of the LSTM.

- The input size of the two linear layers for the actor and critic respectively (i.e. the policy and value heads).

Similar to prior work (Kaplan et al., 2020; Hoffmann et al., 2022), we found $6ND$ to be a good approximation for the number of FLOPs used based on model size $N$ and number of samples $D$. This is because we found there to be about $2ND$ FLOPs in the forward pass, and we assume the backward pass takes about the twice the number of FLOPs from the forward pass.

For the RL experiments, there is a slight change in the way we count FLOPs, which is that we count every forward pass number of FLOPs from the learner twice, since there is a corresponding forward pass from an actor. Hence, for RL our formula becomes $8ND$.

---

[7]https://github.com/Miffyli/nle-sample-factory-baseline

# F TRAINING DETAILS

## F.1 ATARI

We list the hyperparameters for all our BC experiments in Atari in Table 4a, and the ones to train expert policies for each Atari game in Table 4b. To train these expert policies, we used the Stable Baselines3 (Raffin et al., 2021) implementation of PPO (Schulman et al., 2017). We use Adam (Kingma & Ba, 2014) as our optimizer for both settings. All training experiments were done on NVIDIA GPUs (a mix of GeForce RTX 3090, GeForce RTX 2080 Ti, RTX A5000, and RTX A6000) and took about 1 - 2 days depending on the game and FLOP budget.

Table 4: **Hyperparameters for all experiments in Atari.** We list the hyperparameters for all our BC experiments (**a**) as well as the ones used to train the PPO expert agent for each game (**b**).

| Hyperparameters | Value |
|---|---|
| Learning rate | 0.0001 |
| Batch size | $128 \times 1$ (1 GPU) |
| Unroll length | 32 |
| Data workers | 30 |

(a) BC hyperparameters

| Hyperparameter | Value |
|---|---|
| Learning rate | $2.5e^{-4}$ |
| Learner batch size | 256 |
| Entropy cost | 0.01 |
| Baseline cost | 0.5 |
| Total timesteps | $5e^7$ |
| Discount factor | 0.99 |
| GAE lambda | 0.95 |
| Clip range | 0.2 |

(b) PPO hyperparameters

## F.2 NETHACK

We use Adam (Kingma & Ba, 2014) as our optimizer for all BC experiments. For RL, we use RMSprop. Please find all hyperparameters for both settings in Table 5, all of which were manually tuned or taken from prior work. All experiments were run on NVIDIA V100 32GB or A100 40GB GPUs, and took up to 4 days to run, depending on the FLOP budget. The one exception to this is our forecasted BC model which consisted of 30M parameters and was run on 115B samples. This took approximately 11 days, and was partially run on a A100 GPU, and continued on a V100 GPU.

The NLD-AA dataset (Hambro et al., 2022b) is released under the NetHack General Public License and can be found at `https://github.com/dungeonsdatasubmission/dungeonsdata-neurips2022`.

Table 5: **Hyperparameters for all experiments in NetHack.** We list the hyperparameters for all our BC experiments (**a**) as well as the ones for our RL experiments (**b**).

| Hyperparameters | Value |
|---|---|
| Learning rate | 0.0001 |
| Batch size | $128 \times 8$ (8 GPUs) |
| Unroll length | 80 |
| Ttyrec workers | 30 |

(a) BC hyperparameters

| Hyperparameter | Value |
|---|---|
| Learning rate | 0.0002 |
| Learner batch size | 32 |
| Unroll length | 80 |
| Entropy cost | 0.001 |
| Baseline cost | 0.5 |
| Maximum episode steps | 5000 |
| Reward normalization | yes |
| Reward clipping | none |
| Number of actors | 90 |
| Discount factor | 0.99 |

(b) IMPALA hyperparameters

## G    DELTA METHOD DERIVATION

Following standard linear regression assumptions, we have that $\sqrt{n}(\hat{\boldsymbol{\beta}} - \boldsymbol{\beta}) \xrightarrow{D} \mathcal{N}(0, \Sigma)$, where $\hat{\boldsymbol{\beta}}$ is a vector containing all regression coefficients, i.e.

$$\hat{\boldsymbol{\beta}} = \langle \hat{\beta}_0, \hat{\beta}_N, \hat{\beta}_D, \hat{\beta}_{N^2}, \hat{\beta}_{ND}, \hat{\beta}_{D^2} \rangle$$

We also rewrite $\alpha$ from Equation 5 more explicitly as a function $h$:

$$\alpha = h(\hat{\boldsymbol{\beta}}) = \frac{2\beta_{D^2} - \beta_{ND}}{2\beta_{D^2} - 2\beta_{ND} + 2\beta_{N^2}}$$

Approximating $h(\hat{\boldsymbol{\beta}})$ with a first-order Taylor expansion around $\boldsymbol{\beta}$ we get:

$$h(\hat{\boldsymbol{\beta}}) \approx h(\boldsymbol{\beta}) + \nabla h(\boldsymbol{\beta})^T \cdot (\hat{\boldsymbol{\beta}} - \boldsymbol{\beta})$$

Computing the variance of $h(\hat{\boldsymbol{\beta}})$ then gives[8]:

$$\begin{aligned}
\text{Var}[h(\hat{\boldsymbol{\beta}})] &= \text{Var}[h(\boldsymbol{\beta}) + \nabla h(\boldsymbol{\beta})^T \cdot (\hat{\boldsymbol{\beta}} - \boldsymbol{\beta})] \\
&= \text{Var}[h(\boldsymbol{\beta}) + \nabla h(\boldsymbol{\beta})^T \cdot \hat{\boldsymbol{\beta}} - \nabla h(\boldsymbol{\beta})^T \cdot \boldsymbol{\beta}] \\
&= \text{Var}[\nabla h(\boldsymbol{\beta})^T \cdot \hat{\boldsymbol{\beta}}] \\
&= \nabla h(\boldsymbol{\beta})^T \cdot \text{Cov}(\hat{\boldsymbol{\beta}}) \cdot \nabla h(\boldsymbol{\beta}) \\
&= \nabla h(\boldsymbol{\beta})^T \frac{\Sigma}{n} \nabla h(\boldsymbol{\beta})
\end{aligned}$$

Now the Standard Error (SE) can be found as

$$\text{SE} = \sqrt{\nabla h(\boldsymbol{\beta})^T \frac{\Sigma}{n} \nabla h(\boldsymbol{\beta})}.$$

Finally, we approximate the SE above by using $\nabla h(\hat{\boldsymbol{\beta}})$ instead of $\nabla h(\boldsymbol{\beta})$, giving the following confidence interval:

$$\text{CI} = h(\hat{\boldsymbol{\beta}}) \pm 1.96 \cdot \text{SE}.$$

## H    RL SCALING LAWS IN NETHACK

Table 6: Model and data size predictions for RL scaling laws in NetHack.

| RL - Avg. Human (127k) | | |
| --- | --- | --- |
| 1. IsoFLOP profiles | 4.4B | 13.2T |
| 2. Parametric fit | 67B | 0.93T |

We train LSTM-based agents on the `NetHackScore-v0` environment. `NetHackScore-v0` features the full game of NetHack but has a reduced action space, starting character fixed to human monk, and automatic menu skipping. We also experimented with the `NetHackChallenge-v0` environment but found the results too noisy at the FLOP budgets we were able to run. However, we expect similar results will hold for this environment at larger FLOP budgets.

---

[8]We follow a very similar derivation as in `https://en.wikipedia.org/wiki/Delta_method#Multivariate_delta_method`

**IsoFLOP profiles.**   We train 9 different model sizes ranging from 100k to 50M using IMPALA (Espeholt et al., 2018), each with a FLOP budget ranging from $1e13$ to $1e17$. For each of these models, we evaluate the model at the end of training by rolling it out 1k times in the environment and reporting the average return. While learning curves in RL tend to have high variance, we generally still find that compute-optimal models should increase both the number of parameters and number of environment interactions as the FLOP budgets are scaled up (see Figure 4). We also find that the NetHack game score varies smoothly with FLOPs and hence can be seen as a *natural performance metric* (Hilton et al., 2023), as discussed in more detail in section 6. We again follow a similar procedure as in subsection 4.1 resulting in power laws as listed in Equation 6. We find $\alpha = 0.43$, $\beta = 0.56$, and $\gamma = 0.32$.

**Parametric fit**   We take the functional form in Equation 3, and replace loss with mean return. We can then solve the same constrained optimization problem resulting in the exact same expressions as found in Equation 5 (the denominator of 6 is replaced with 8 due to a slight difference in FLOP counting for RL, see Appendix E). After fitting, we find $\alpha = 0.60$ and $\beta = 0.40$. Note we dropped the low flop budgets when performing this regression, as we found this greatly improved the fit.

**Forecasting human performance.**   Hambro et al. (2022b) report that average human performance is around 127k. Based on the two approaches discussed above, we forecast the compute requirements for training an RL agent from scratch to achieve human-level performance on NetHack, listed in Table 6. For the isoFLOP profile approach, we first use Figure 4b to solve for $C_{127k}$. Then we plug this into the power laws from Figure 4c and Figure 4d. For the parametric fit, we instead plug $C_{127k}$ into the power laws from Equation 5 with the correct $\alpha$ and $\beta$ from above, where the denominator of 6 is replaced with 8 as mentioned earlier. In Table 6, we find the parametric fit to put significantly more emphasis on model size, which could be possible due to dropping of the low FLOP budgets (optimal model size tends to shift more clearly in larger FLOP budgets). Due to computational constraints, we leave testing the limits of this prediction to future work. Finally, we perform rolling time series cross-validation to evaluate one-step ahead forecasting performance, as described in Appendix J.

**RL with pretraining.**   All our scaling law results on the RL side in this paper are with policies trained *from scratch*. However, some of the most promising neural methods for NetHack and other domains leverage a pre-trained (e.g. through imitation learning) policy that is then finetuned with RL. It would be very interesting to analyze the scaling behaviors for these kind of kickstarted policies, and see whether they scale differently than the ones trained from scratch. We leave this to future work.

## I   ATARI FULL RESULTS

Figure 6, Figure 7, Figure 8, and  Figure 9 list the full set of Atari results with respect to cross entropy loss. Figure 10, Figure 11, Figure 12, and Figure 13 list the full set of Atari results with respect to environment return. Finally, Figure 14 lists the full set of Atari results relating environment return and optimal loss. Note that for the return results, we can see that Space Invaders is the only Atari game where didn't run high enough FLOP budgets to reach expert performance.

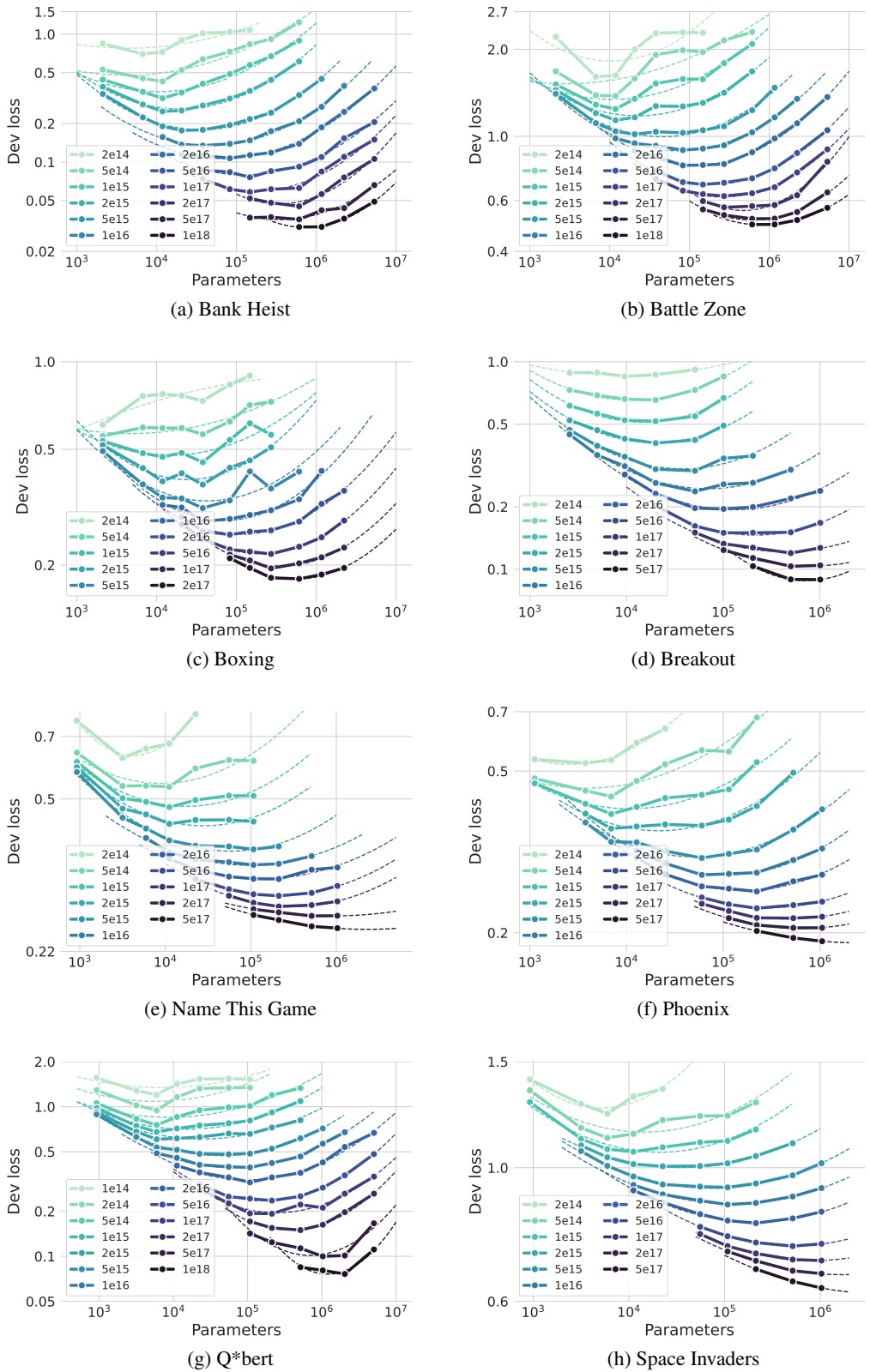

Figure 6: **BC isoFLOP curves.** Full results for all Atari games.

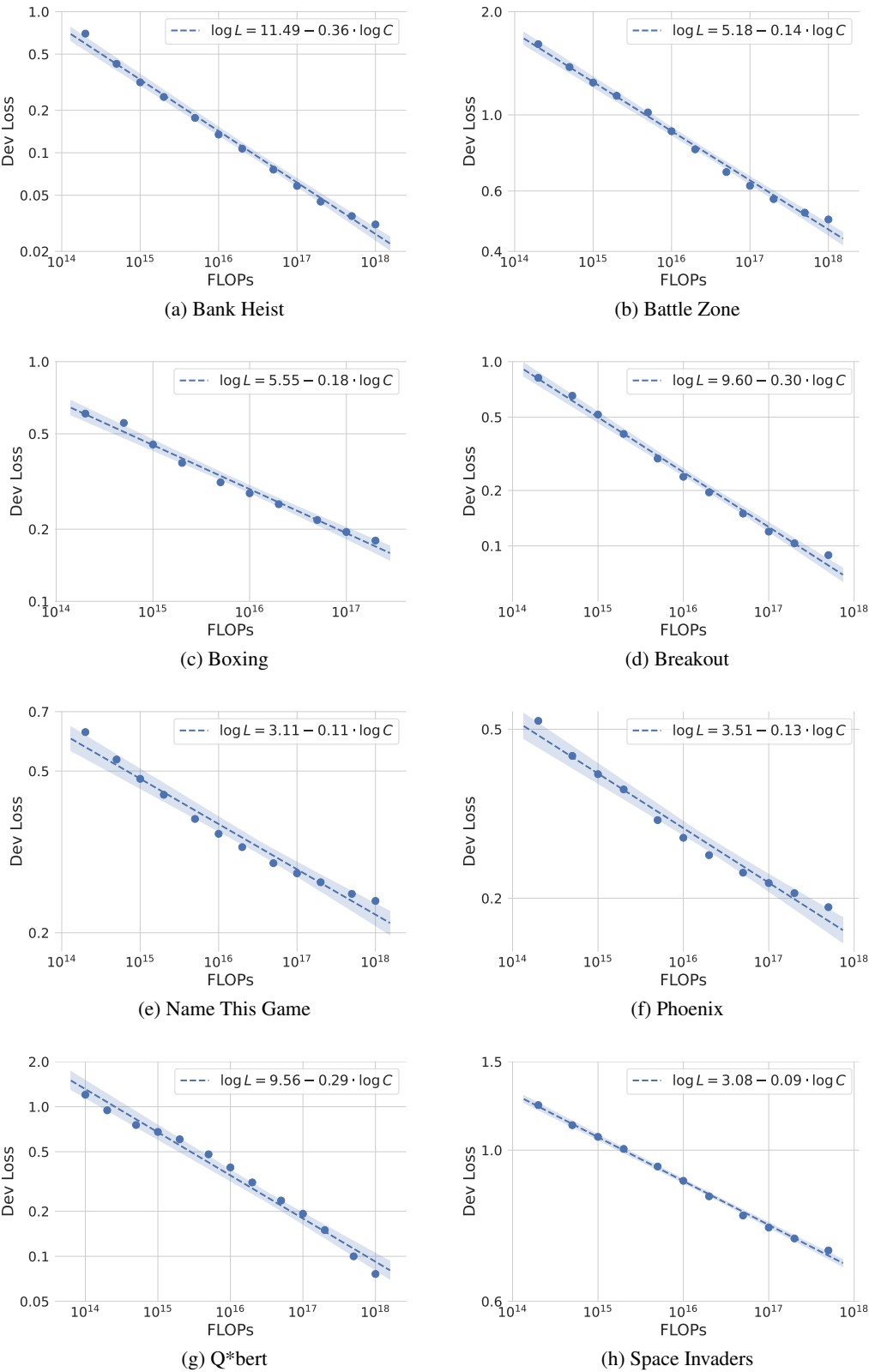

Figure 7: **BC optimal loss vs. FLOPs curves.** Full results for all Atari games.

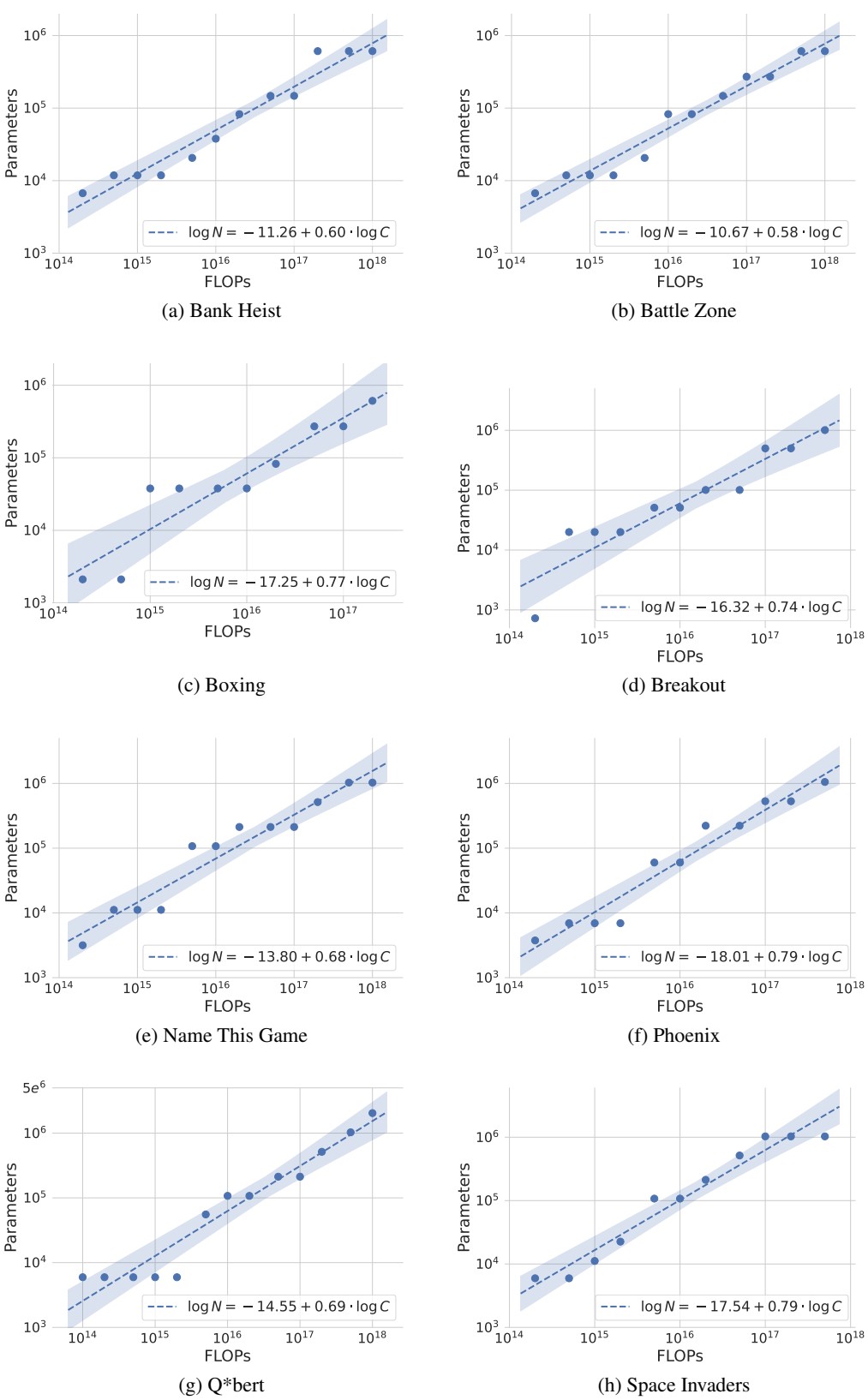

Figure 8: **BC optimal parameters vs. FLOPs curves.** Full results for all Atari games.

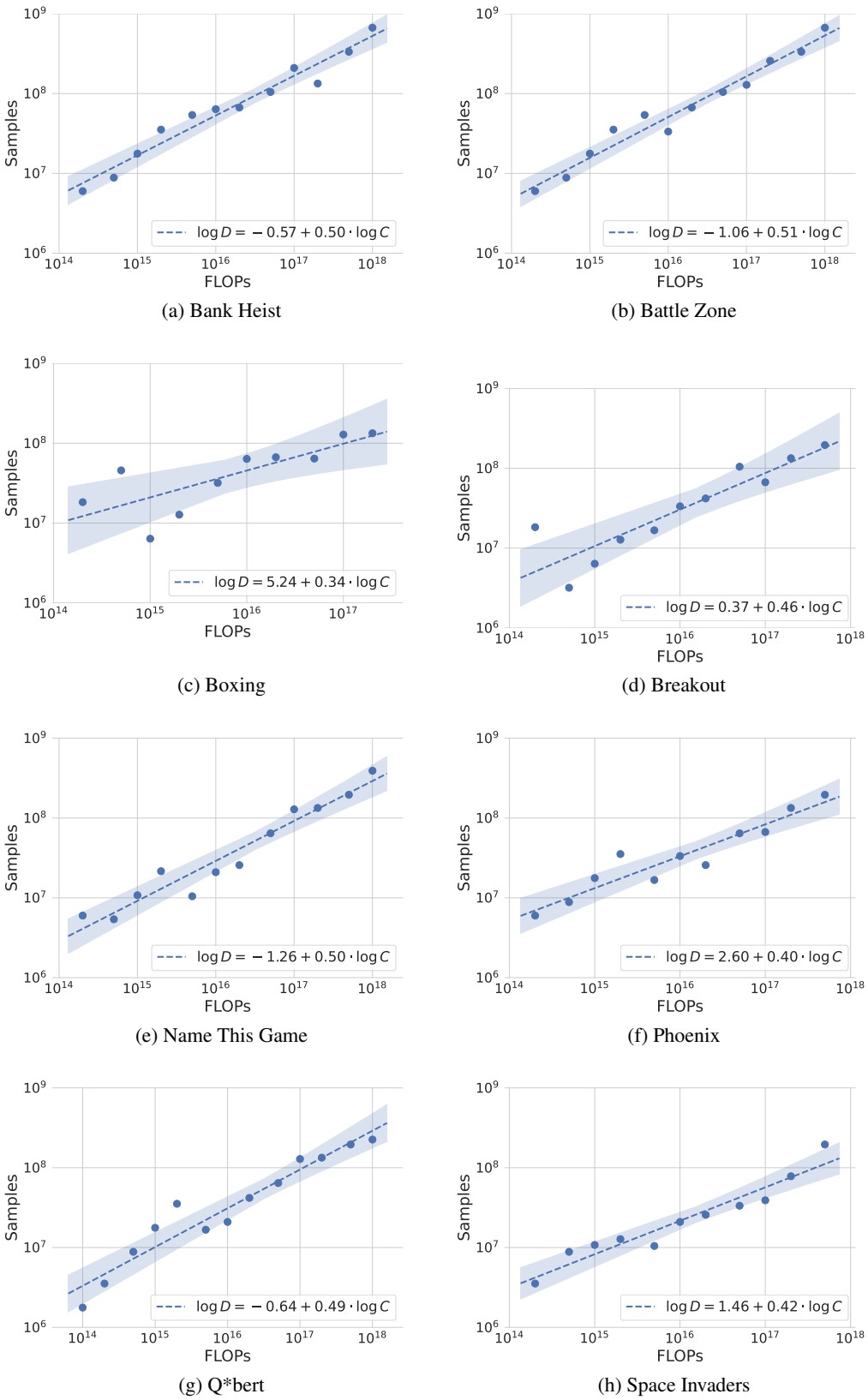

Figure 9: **BC optimal samples vs. FLOPs curves.** Full results for all Atari games.

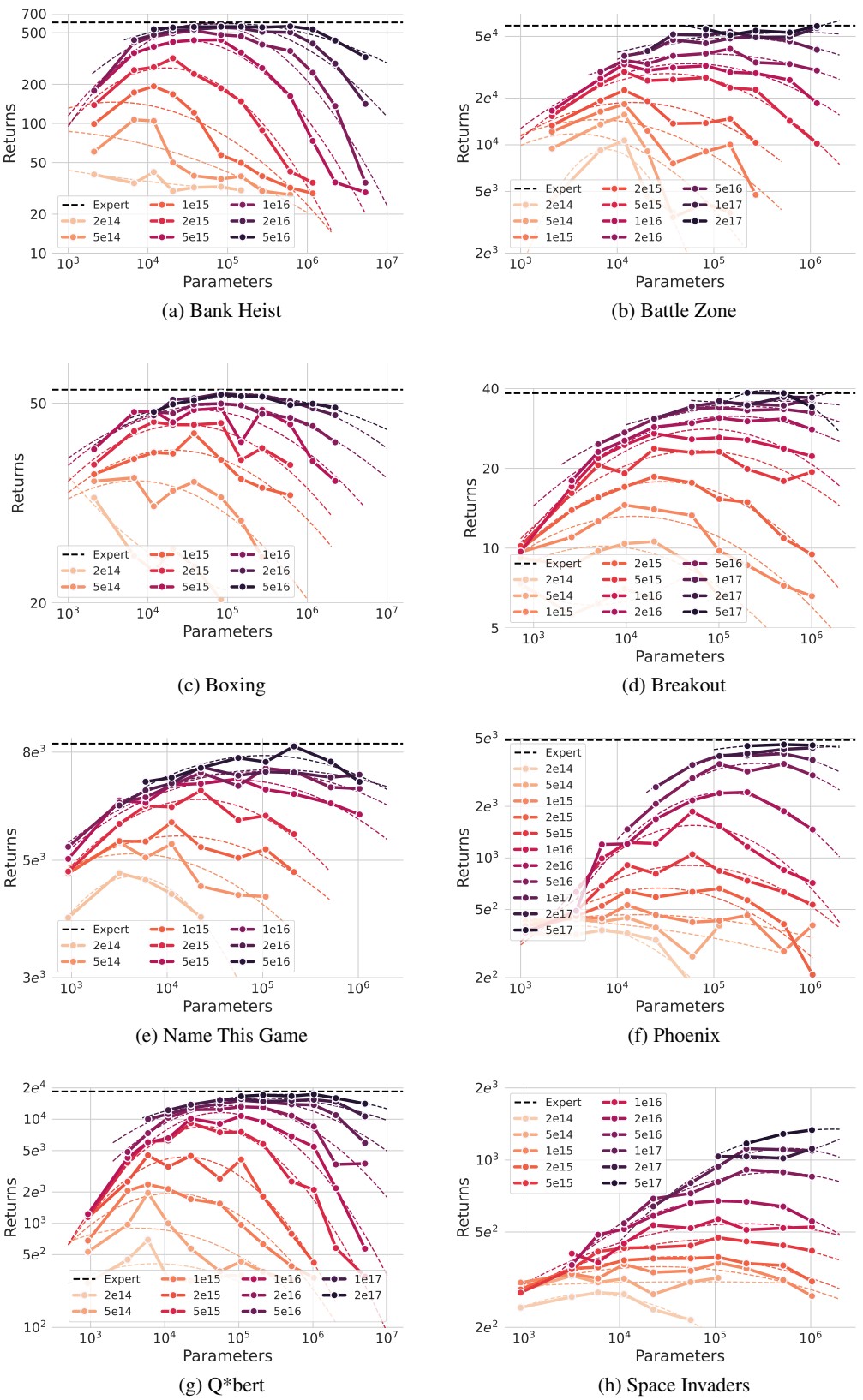

Figure 10: **BC return isoFLOP curves.** Full results for all Atari games.

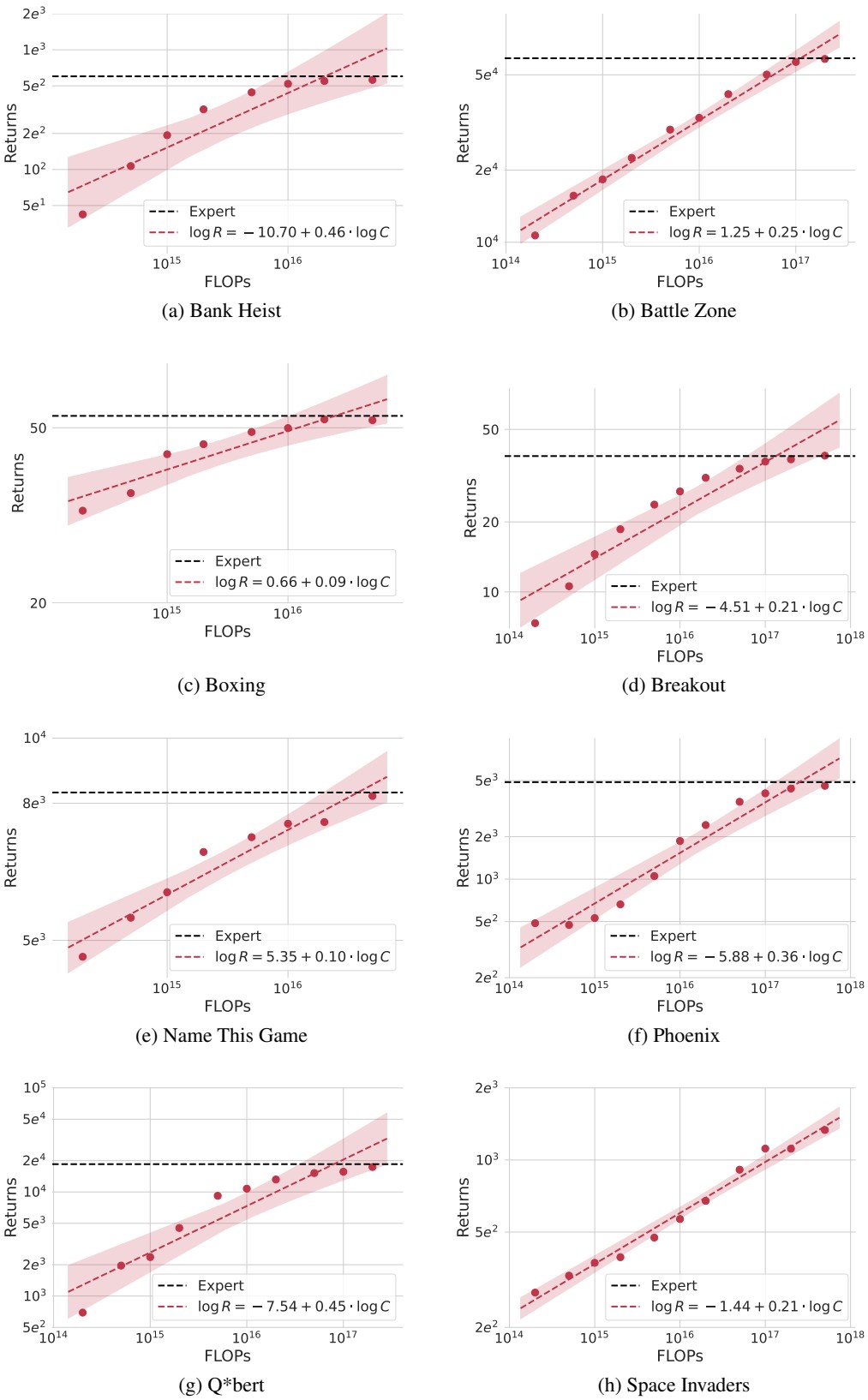

Figure 11: **BC optimal returns vs. FLOPs curves.** Full results for all Atari games.

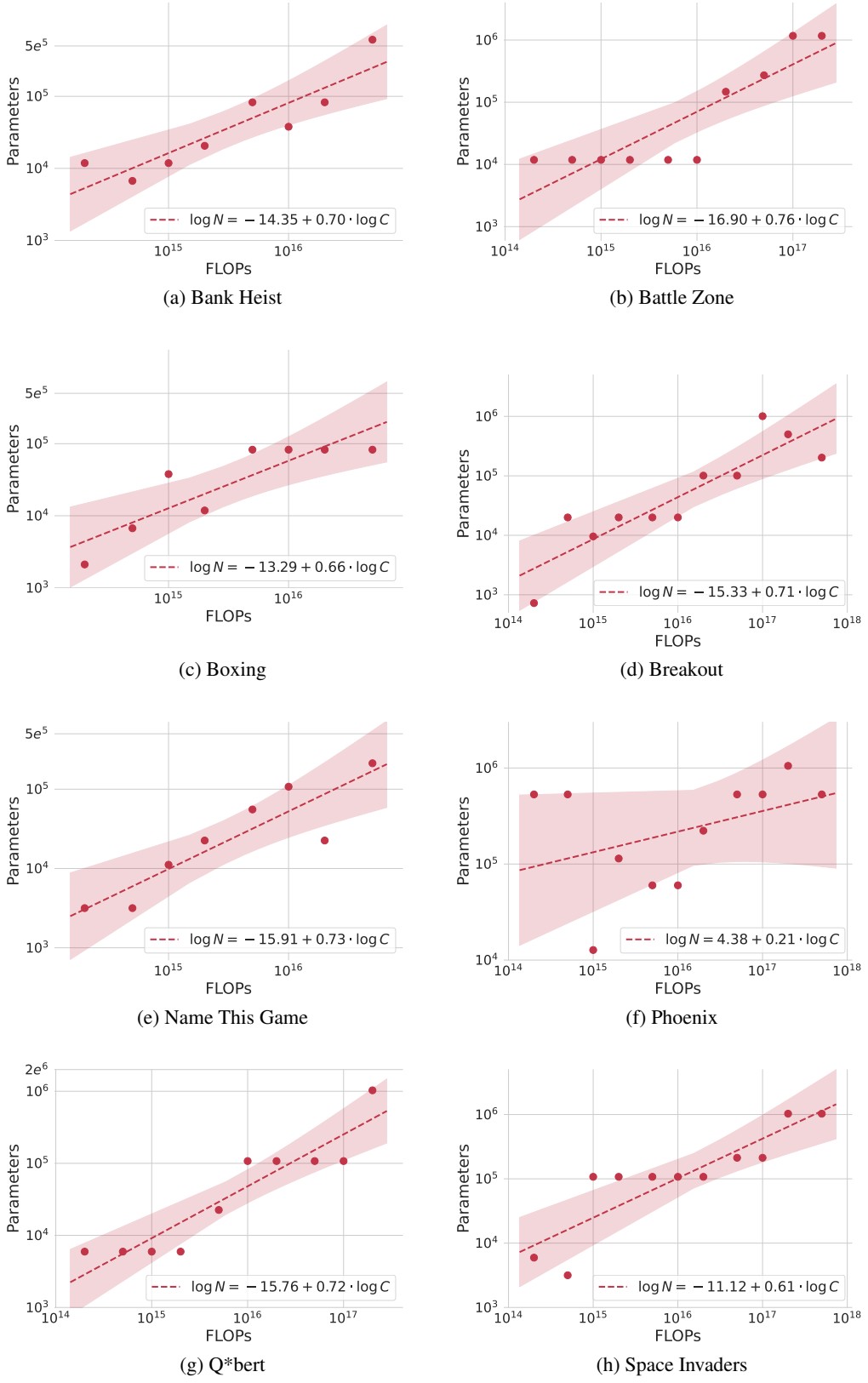

Figure 12: **BC return-optimal parameters vs. FLOPs curves.** Full results for all Atari games.

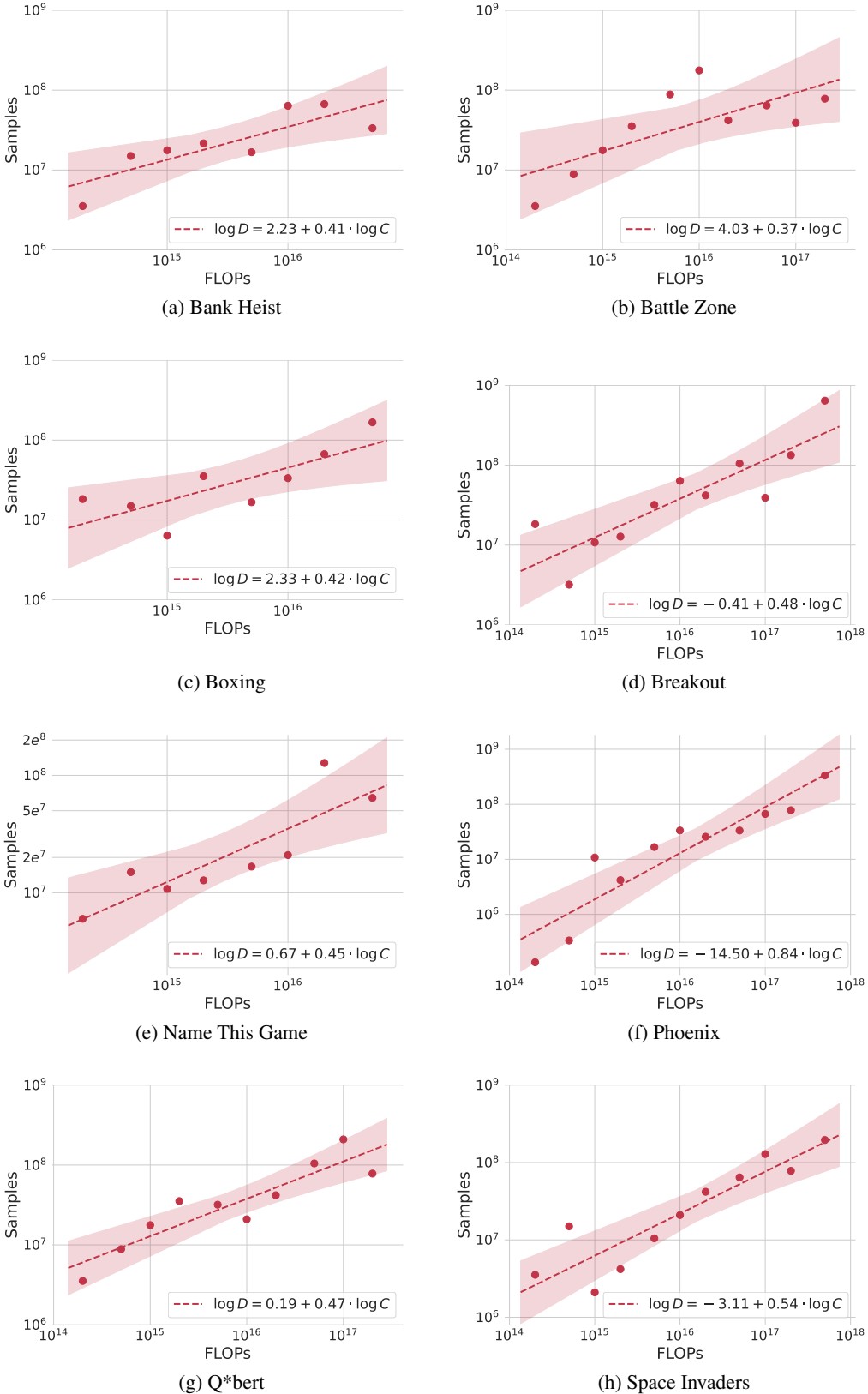

Figure 13: **BC return-optimal samples vs. FLOPs curves.** Full results for all Atari games.

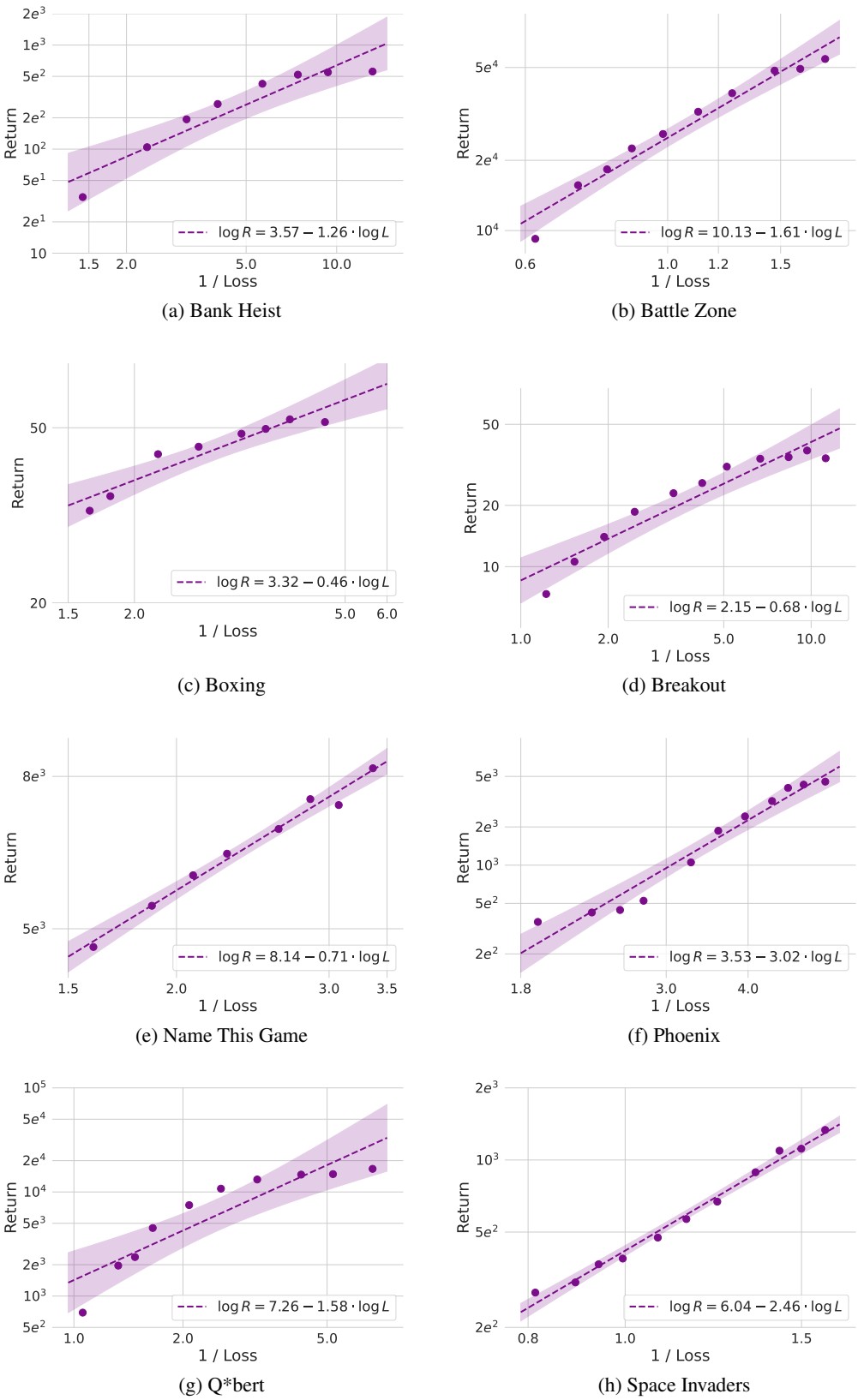

Figure 14: **BC return vs. optimal loss.** Full results for all Atari games.

## J   NetHack Forecasting Cross-validation

To measure forecasting performance of all power law regressions done for NetHack, we leverage a rolling time series cross-validation. Specifically, for every regression, we initially take the first 6 points (corresponding to the first 6 FLOP budgets) to be our training set to fit the regression, and then compute the RMSE on the 7th point. Then, we include the 7th point in our training set and evaluate on the 8th point. This process keeps going until all but the last point is included, after which we average all resulting RMSEs. Note that this process ensures we always evaluate on *future* FLOP budgets. Results can be found in Table 7.

To provide further insight into our time-series-based cross-validation results, we also visualize the progression of the $\beta_0$ and $\beta_1$ parameters in Figure 15, Figure 16, Figure 17, and Figure 18.

Table 7: **Cross-validation RMSEs.** We use a time series rolling cross-validation to compute the average Root Mean Squared Error (RMSE) for all regressions done in the paper.

| Regressions | Dev RMSE |
|---|---|
| Loss minima vs. FLOPs ( Figure 1b, left) | $7.7e^{-2}$ |
| Loss-optimal parameters vs. FLOPs ( Figure 1c, left) | $4.3e^5$ |
| Loss-optimal samples vs. FLOPs ( Figure 1d, left) | $3.8e^9$ |
| BC maximal returns vs. FLOPs ( Figure 2b, left) | $3.9e^2$ |
| BC return-optimal parameters vs. FLOPs ( Figure 2c, left) | $1.6e^6$ |
| BC return-optimal samples vs. FLOPs ( Figure 2d, left) | $1.7e^{10}$ |
| BC returns vs. loss minima ( Figure 3a, left) | $1.2e^2$ |
| RL maximal returns vs. FLOPs ( Figure 4b, left) | $1.4e^2$ |
| RL return-optimal parameters vs. FLOPs ( Figure 4c, left) | $2.9e^6$ |
| RL return-optimal samples vs. FLOPs ( Figure 4d, left) | $1.6e^9$ |

## K   Observation & action space

For Atari, the observation space consists of an 84 x 84 image of pixel values between 0 and 255. The action space is discrete and varies per game. See below for the size of the action space per game:

- Battle Zone: 18
- Q*bert: 6
- Bank Heist: 18
- Boxing: 18
- Breakout: 4
- Name This Game: 6
- Phoenix: 8
- Space Invaders: 6

For NetHack, the observation space consists of 24 x 80 ASCII character and color grids (one for each), along with the cursor position (consisting of 2 coordinates). The action space is discrete and consists of 121 actions for the full game (i.e. `NetHackChallenge-v0`) and 23 actions for the simplified version (i.e. `NetHackScore-v0`).

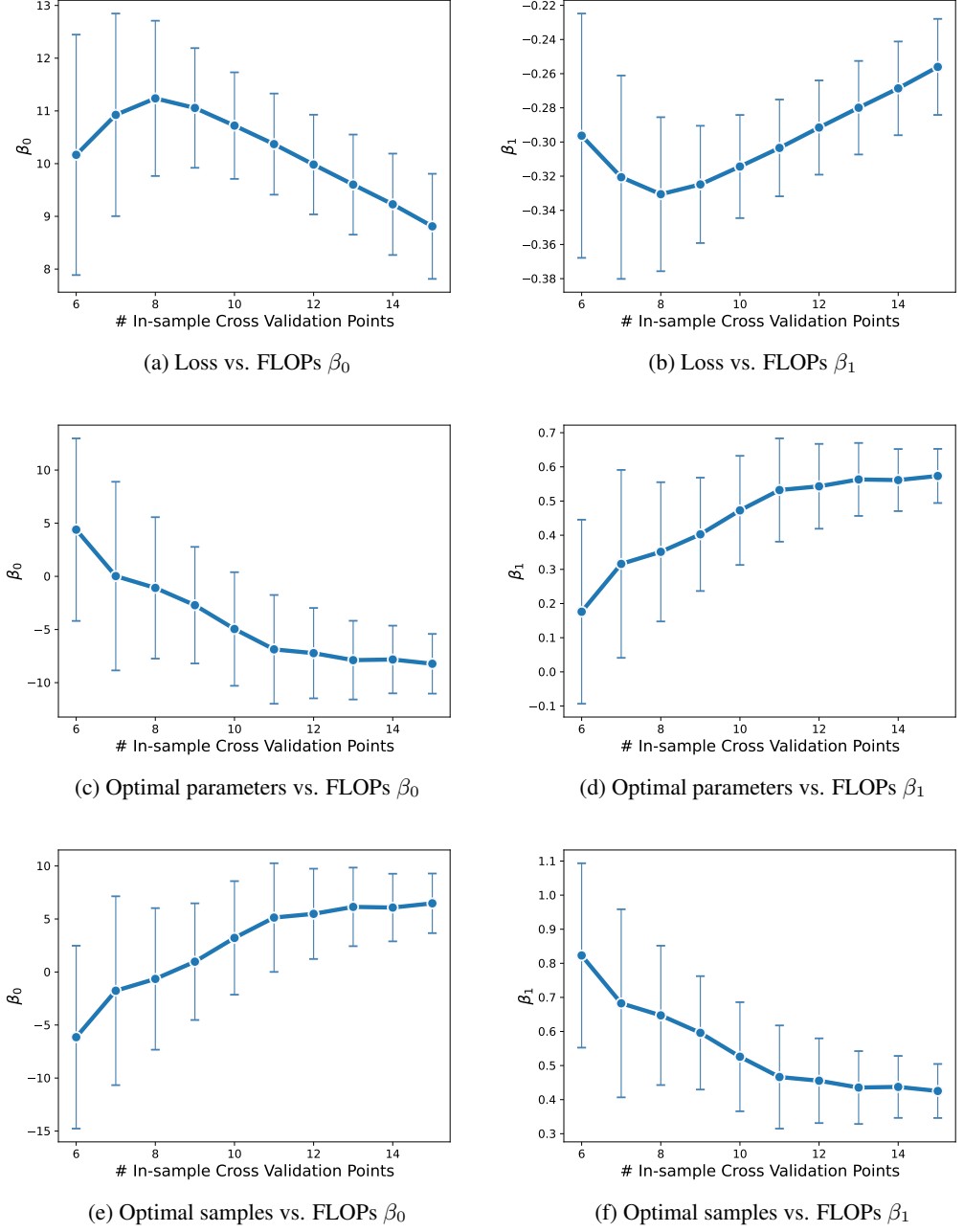

(a) Loss vs. FLOPs $\beta_0$

(b) Loss vs. FLOPs $\beta_1$

(c) Optimal parameters vs. FLOPs $\beta_0$

(d) Optimal parameters vs. FLOPs $\beta_1$

(e) Optimal samples vs. FLOPs $\beta_0$

(f) Optimal samples vs. FLOPs $\beta_1$

Figure 15: **BC loss cross validation progression.** We leverage a time-series-based rolling cross-validation going from early to later FLOP budgets, and visualize the progression of the $\beta_0$ and $\beta_1$ coefficients.

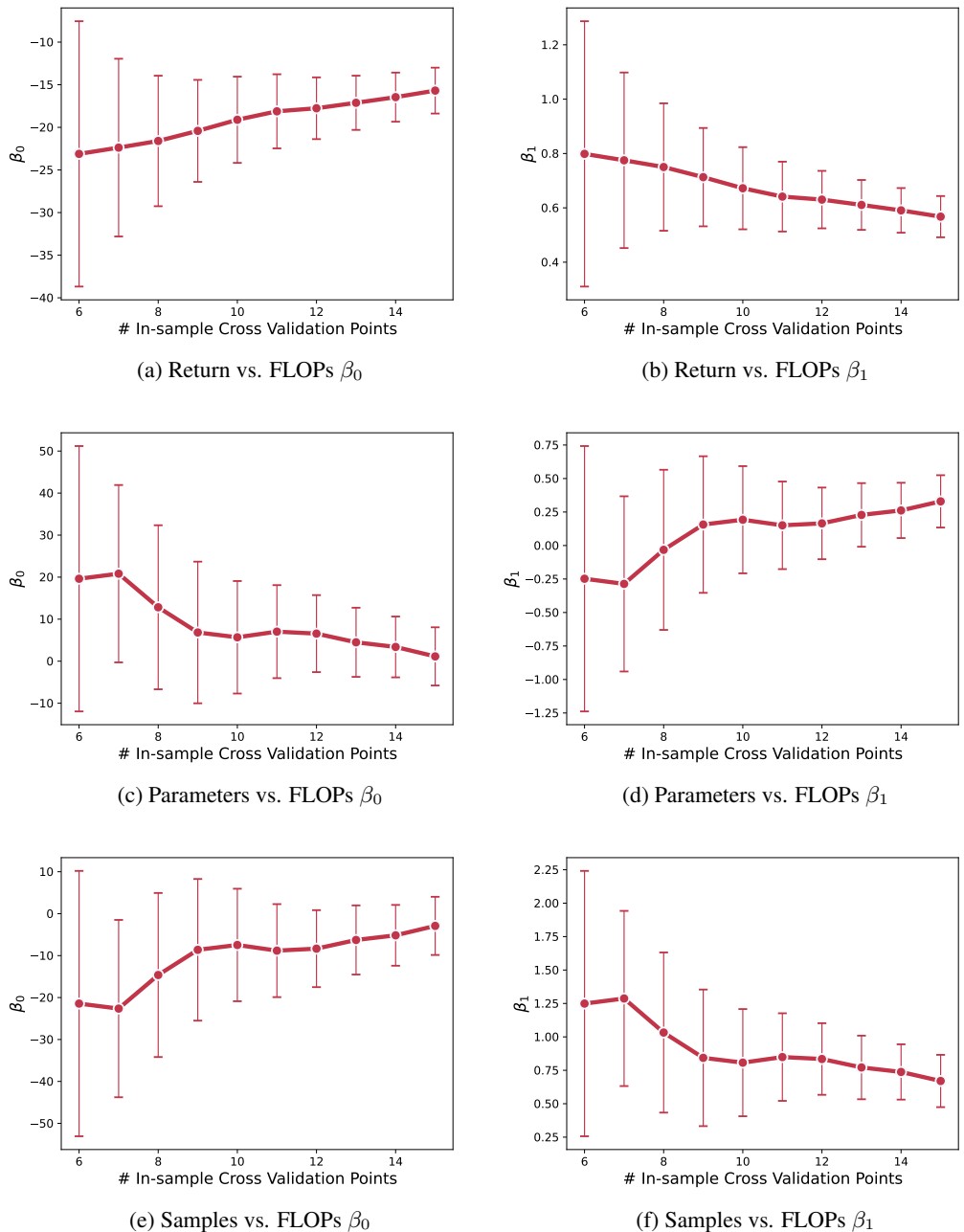

Figure 16: **BC return cross validation progression.** We leverage a time-series-based rolling cross-validation going from early to later FLOP budgets, and visualize the progression of the $\beta_0$ and $\beta_1$ coefficients.

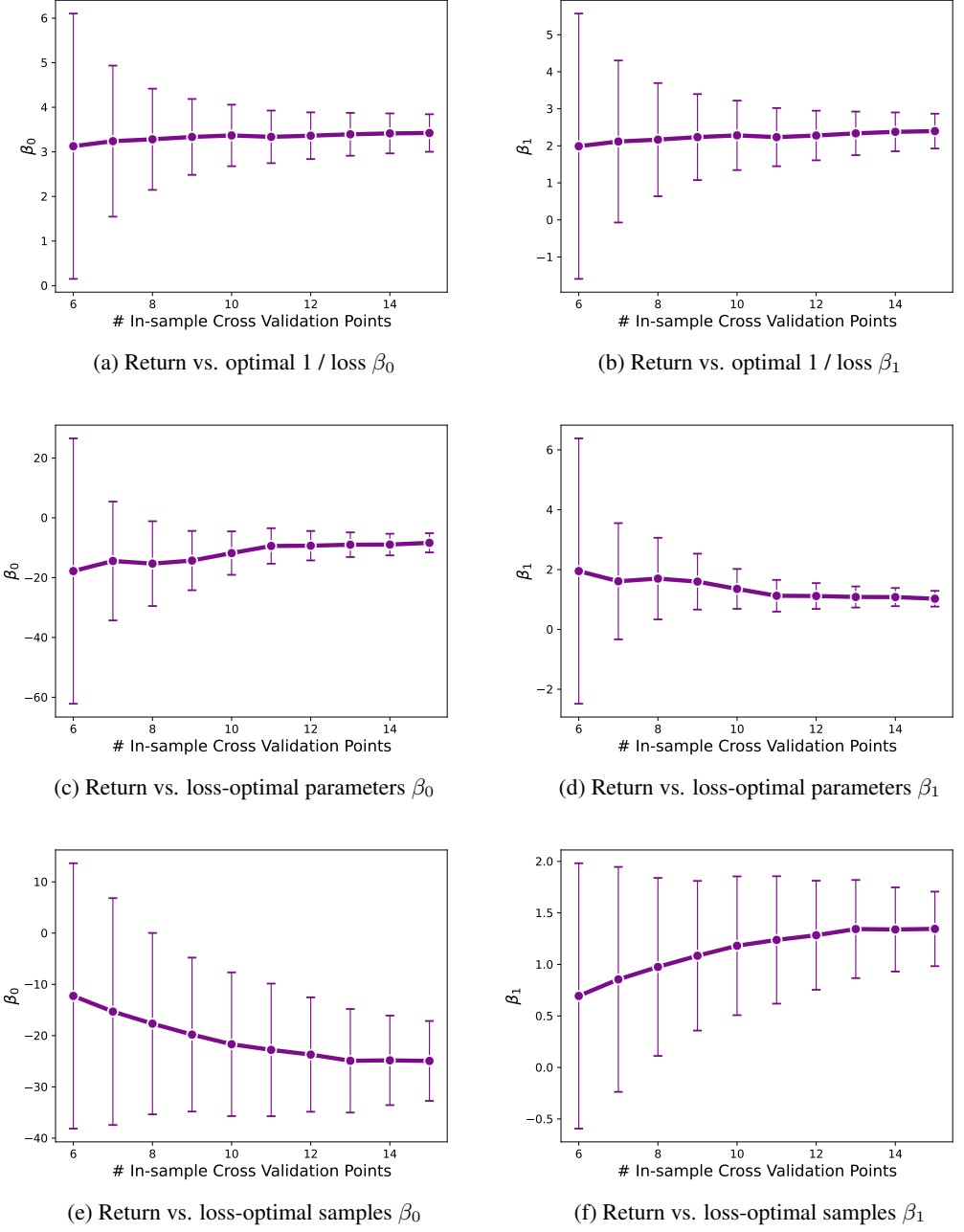

(a) Return vs. optimal 1 / loss $\beta_0$

(b) Return vs. optimal 1 / loss $\beta_1$

(c) Return vs. loss-optimal parameters $\beta_0$

(d) Return vs. loss-optimal parameters $\beta_1$

(e) Return vs. loss-optimal samples $\beta_0$

(f) Return vs. loss-optimal samples $\beta_1$

Figure 17: **BC return-loss relation cross validation progression.** We leverage a time-series-based rolling cross-validation going from early to later FLOP budgets, and visualize the progression of the $\beta_0$ and $\beta_1$ coefficients.

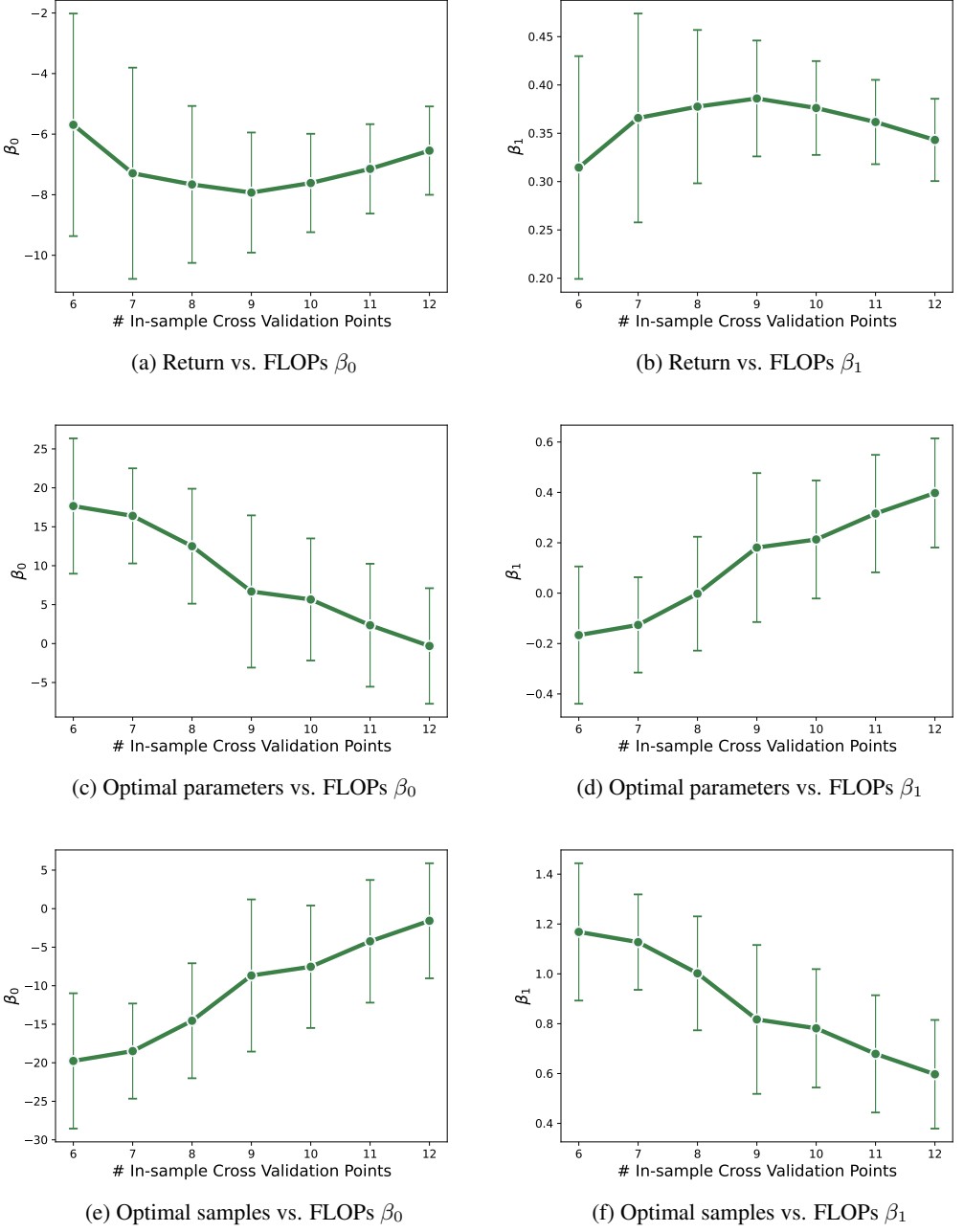

Figure 18: **RL return cross validation progression.** We leverage a time-series-based rolling cross-validation going from early to later FLOP budgets, and visualize the progression of the $\beta_0$ and $\beta_1$ coefficients.

