# OpenReview forum: "Scaling Laws for Imitation Learning in Single-Agent Games"
_ICLR.cc/2024/Conference — Submitted to ICLR 2024_

### Official Review · Reviewer_fMKL · 2023-10-15

**Soundness:** 3 good
**Presentation:** 3 good
**Contribution:** 4 excellent
**Rating:** 8
**Confidence:** 3

**Summary:**

This paper is an empirical study on how scaling up compute budget (FLOPs) would affect imitation learning (IL) loss and mean return in single-agent games, and find that both metrics scales smoothly with the compute budget. Based on the findings, the work also forecasts and trains agents with IL on NetHack, which largely outperforms state-of-the-art. The work also studies the scaling law of RL with IMPALA.

**Strengths:**

1. The problem studied in this paper is meaningful for the RL community. While the deep learning community "scales up" in recent years with ever-growing number of hyperparameters and computational resources, many RL/IL works still use simple MLP over relatively low-dimensional environments such as mujoco locomotion. Such work is a complement to the current RL/IL community.

2. The empirical study presented in this paper is thorough with an extensive amount of experiments, and the result clearly supports the claim that the performance of IL scales smoothly with computational budget and number of parameters.

3. The paper is well-written, with a clear discussion on limitations (reward as metrics could be non-smooth, hyperparameters are not tuned for each profile), possible future directions, ethics statement and reproducibility statement, and implementation details carefully listed. The core message of scaling law is also well-conveyed.

**Weaknesses:**

1. While scaling law of behavior cloning (and RL) is useful and important for the RL community beyond standard practice of MLPs, its use in real-life applications is still limited. In order for behavior cloning to learn well, the dataset must be of good quality since it has no selection over the dataset; however, expert dataset is usually costly in real-life and of small size.

2. The results are tested on Atari benchmarks, which are discrete (it would be better if the action and state space of NetHack and Atari could be introduced in the appendix). It would be better if the results on environments with continuous action space is shown.

3. "isoFLOP" is not formally introduced in the paper.

4. typo: "consistenly" -> "consistently" (introduction, paragraph 2, line 5)

**Questions:**

As the authors mention that they exclude Double Dunk because it is too simple to learn, I wonder whether the scaling law works for smaller models and simpler environment (e.g. gym/mujoco environments) or it only works for more complicated environments such as Atari and NetHack.

---

> ### Author Response · Authors · 2023-11-16
> **Response to reviewer fMKL**
>
> We thank the reviewer for their valuable feedback, and are happy to hear they feel very positive about it!
>
> > While scaling law of behavior cloning (and RL) is useful and important for the RL community beyond standard practice of MLPs, its use in real-life applications is still limited. In order for behavior cloning to learn well, the dataset must be of good quality since it has no selection over the dataset; however, expert dataset is usually costly in real-life and of small size.
>
> First, we’d like to point out that **there are real-world applications with plenty of expert data where our work could have practical applications.** One example is autonomous driving as pointed out by Rajaraman et al. [7], who refers to the work of Chen et al. [8] where a model-based controller can be used to collect as much data in the simulator as desired. They also note that human experts could be used instead of model-based controllers (which makes sense especially in the context of companies like Tesla or Waymo).
>
> Second, we do agree investigating scaling up methods that aim to relax reliance on expert data by either (1) leveraging additional imperfect data (e.g. Kim et al. [9]) or (2) “stitching” trajectories from suboptimal data to improve behavior (i.e. offline RL) could be an interesting and important direction. **We hope the promising trends in our work can inspire future work to tackle these questions.**
>
> > The results are tested on Atari benchmarks, which are discrete (it would be better if the action and state space of NetHack and Atari could be introduced in the appendix). It would be better if the results on environments with continuous action space is shown.
>
> Thank you for this suggestion! **We have updated the appendix with an overview of the action space for both NetHack and Atari** (see text in magenta in appendix K). We agree environments with continuous action spaces would be good to investigate, but leave this to future work.
>
> > "isoFLOP" is not formally introduced in the paper.
>
> Thank you for pointing this out! **We have updated the pdf to include a definition of “isoFLOP”** (see text in magenta in section 4.1).
>
> > typo: "consistenly" -> "consistently" (introduction, paragraph 2, line 5)
>
> Thanks for catching this! **We have updated the pdf to fix this** (see text in magenta in the introduction).
>
> > As the authors mention that they exclude Double Dunk because it is too simple to learn, I wonder whether the scaling law works for smaller models and simpler environment (e.g. gym/mujoco environments) or it only works for more complicated environments such as Atari and NetHack.
>
> We would suspect the scaling laws still hold up for simpler environments as well (though one may have to consider smaller compute budgets and smaller model sizes). In fact, NetHack is arguably a much more complex environment than Atari, though we see scaling laws in both. However, **note that studying scale might be most interesting in more complex environments** where we can test methods across multiple orders of magnitude of compute budgets and not reach maximal performance right away. This is one reason why we included NetHack: it is a completely unsolved environment where studying scale could tell us if existing methods like IL can keep giving us improvements, or whether there are fundamental limits to those methods even with scale.

---

> > ### Comment · Reviewer_fMKL · 2023-11-17
> > **Response to Rebuttal**
> >
> > Thanks for the authors' detailed response; I think my problems are addressed and will keep my current score.

---

> > > ### Author Response · Authors · 2023-11-17
> > > **Thank you**
> > >
> > > We would like to thank the reviewer again for their feedback as well as their active engagement in the rebuttal!

---

### Official Review · Reviewer_aGsp · 2023-10-28

**Soundness:** 3 good
**Presentation:** 3 good
**Contribution:** 2 fair
**Rating:** 6
**Confidence:** 4

**Summary:**

This paper explores the scaling laws for single-agent imitation learning (IL). In examining the Atari and Nethack games, it discovers that IL loss and mean return scale smoothly with the compute budget (FLOPs) and exhibit a strong correlation, culminating in power laws for training compute-optimal IL agents.

**Strengths:**

- Impressive empirical results from applying scaling laws to the challenging game of NetHack.
- This paper is well-written and generally easy to follow.


====

After rebuttal, I decided to increase my score from 5 to 6

**Weaknesses:**

- Missing related work

The scaling laws bear a close relation to the sample complexity theory as outlined in [1, 2, 3], which also posits that performance can scale up with data size and model capacity. The sample complexity serves as a general framework applicable to all architectures and datasets. This relationship should be acknowledged in the main text.

[1] Rajaraman, Nived, et al. "Toward the fundamental limits of imitation learning." *Advances in Neural Information Processing Systems* 33 (2020): 2914-2924.

[2] Xu, Tian, et al. "Error bounds of imitating policies and environments." *Advances in Neural Information Processing Systems* 33 (2020): 15737-15749.

[3] Rajaraman, Nived, et al. "On the value of interaction and function approximation in imitation learning." *Advances in Neural Information Processing Systems* 34 (2021): 1325-1336.

- Limited impact

In many domains such as robotics and healthcare, acquiring additional expert data is costly, even if resources are available for training robust models. This scenario contrasts with NLP applications where collecting high-quality data is relatively inexpensive. Consequently, the practical impact of this paper might be limited. The reviewer suggests that exploring scaling laws in the setup referred to as IL with supplementary and imperfect data, as discussed in [4, 5, 6], could be more valuable since in such cases, supplementary data is more affordable.

[4] Kim, Geon-Hyeong, et al. "Demodice: Offline imitation learning with supplementary imperfect demonstrations." *International Conference on Learning Representations*. 2021.

[5] Wang, Yunke, et al. "Learning to weight imperfect demonstrations." *International Conference on Machine Learning*. PMLR, 2021.

[6] Li, Ziniu, et al. "Theoretical Analysis of Offline Imitation With Supplementary Dataset." *arXiv preprint arXiv:2301.11687* (2023).

**Questions:**

The points in the second column of Figure 2(c) and Figure 4(c) do not exhibit a clear trend or a good fit. Could this paper provide an explanation?

---

> ### Author Response · Authors · 2023-11-15
> **Response to reviewer aGsp**
>
> We thank the reviewer for their valuable feedback.
>
> > The scaling laws bear a close relation to the sample complexity theory as outlined in [1, 2, 3], which also posits that performance can scale up with data size and model capacity. The sample complexity serves as a general framework applicable to all architectures and datasets. This relationship should be acknowledged in the main text.
>
> Thank you for pointing us to these works! While these works provide great insight into various error bounds for imitation learning in several settings, we did not find any results exactly relating sample complexity with scaling laws. Is the reviewer mainly talking about the `N` variable in these bounds, which we could interpret as the dataset size and which (if increased) could improve the suboptimality bounds? Or is there a different relationship the reviewer is referring to here?
>
> **Please let us know, since we would be happy to update the pdf once this relationship is clarified!**
>
> > In many domains such as robotics and healthcare, acquiring additional expert data is costly, even if resources are available for training robust models. This scenario contrasts with NLP applications where collecting high-quality data is relatively inexpensive. Consequently, the practical impact of this paper might be limited. The reviewer suggests that exploring scaling laws in the setup referred to as IL with supplementary and imperfect data, as discussed in [4, 5, 6], could be more valuable since in such cases, supplementary data is more affordable.
>
> First, we’d like to point out that **there *are* real-world applications with plenty of expert data where our work could have practical applications**. One example is autonomous driving as pointed out by Rajaraman et al. [7], who refers to the work of Chen et al. [8] where a model-based controller can be used to collect as much data in the simulator as desired. They also note that human experts could be used instead of model-based controllers (which makes sense especially in the context of companies like Tesla or Waymo).
>
> Second, we would also like to note that **the results in our paper don’t necessarily assume the data comes from an optimal policy**. For example, the NetHack data comes from a rule-based system named AutoAscend, a SOTA model for NetHack as of today. However, this system is very far from solving NetHack, meaning it is a highly suboptimal policy when thinking about solving NetHack. Nevertheless, we observe scaling trends for this policy (though, of course, the upper bound of the scaling laws will be the performance of AutoAscend).
>
> Finally, we agree with the reviewer that exploring whether scaling laws can be found for methods like [4, 5, 6] (references from reviewer, not our global list) that have specific objectives different from BC for leveraging suboptimal data could be super interesting and relevant, but we leave this to future work.
>
> > The points in the second column of Figure 2(c) and Figure 4(c) do not exhibit a clear trend or a good fit. Could this paper provide an explanation?
>
> First, **we checked the confidence interval for the slope coefficient of Figures 2(c) and Figure 4(c) and found they all exclude 0.**
>
> Second, **it’s possible that the fit isn’t as good due to various sources of noise.** Specifically, we kept hyperparameters constant across model sizes (see section 6 on the limitations of this), which could potentially obscure the trends a bit. In addition, we always only trained only one model per point on the isoFLOP profiles. Training multiple seeds per point and averaging might substantially reduce noise in our plots, especially for Figure 4(c) since runs come from training RL policies, which are notoriously high variance.
>
> **We would be happy to run and include updated versions of Figures 2(c) and 4(c) where we tune hyperparameters more extensively and average across a few seeds (say 3 - 5). Would the reviewer find this helpful?** (Note that due to time limitations, we can’t promise to get this done by rebuttal time, though we can try. If the paper would get accepted we can definitely get it by the camera-ready version of the paper.)

---

> > ### Comment · Reviewer_aGsp · 2023-11-19
> >
> > Thank you for your detailed explanation.
> >
> > - The variable \(N\) in the sample complexity theory is highly relevant to scaling laws. In some cases, the number of function classes is also important for the function approximation ability. However, this relation is less clear compared with the sample size \(N\), so this paper does not need to clarify this point. As mentioned, the sample complexity serves as a general framework, which may not be as useful as scaling laws in specific applications. It would be great if this relationship could be clarified.
> >
> > - The concerns about Figure 2(c) and Figure 4(c) are minor. The authors may explore the reasons later.

---

> > > ### Author Response · Authors · 2023-11-19
> > > **Response to Reviewer aGsp**
> > >
> > > > The variable (N) in the sample complexity theory is highly relevant to scaling laws. In some cases, the number of function classes is also important for the function approximation ability. However, this relation is less clear compared with the sample size (N), so this paper does not need to clarify this point. As mentioned, the sample complexity serves as a general framework, which may not be as useful as scaling laws in specific applications. It would be great if this relationship could be clarified.
> > >
> > > We thank the reviewer for the clarification between scaling laws and sample complexity theory. **We have updated our pdf to include this relation in the related work section** (see text in orange).
> > >
> > > We thank the reviewer again for their constructive engagement in the rebuttal, and **we hope our clarifications and modifications allow the reviewer to raise their score**.

---

### Official Review · Reviewer_U8oj · 2023-10-30

**Soundness:** 3 good
**Presentation:** 3 good
**Contribution:** 3 good
**Rating:** 6
**Confidence:** 3

**Summary:**

Recent works in large language models (LLMs) demonstrate that scaling up the model and data size leads to increasingly capable LLMs. Inspired by these works, this paper investigates the scaling law in imitation learning in the domain of single-agent games. In particular, this paper considers the BC method and the games of Atari and NetHack. This paper first considers the metric of MLE loss. The experiment results first show a clear power-law of the optimal model size, data size, and MLE loss with respect to the compute budget. Then this paper considers the metric of policy return and the empirical results show a similar conclusion. Finally, this paper uses the law obtained early to predict the optimal model size and data size required to achieve the expert policy. They train NetHack agents with the predicted optimal model size and data size and find they outperform prior state-of-the-art by 2x in all settings.

**Strengths:**

1. This paper is the first to study the scaling law of imitation learning in the domain of games. This paper confirms that scaling up the model size and data size can result in better agents, which is an important conclusion for IL.
2. This paper conducts systematic experiments to validate the scaling law. The experimental design is sound and the experimental results are convincing.

**Weaknesses:**

1. Given existing works studying the scaling law in LLMs, the novelty of investigating the scaling law of BC in games is limited. As claimed in this paper, LLMs use the same MLE objective as BC. Thus, the only difference between this paper and existing works is the agent domain. As a result, it is not very surprising that the single-agent games domain can exhibit a similar scaling law. A more interesting direction is to investigate the scaling law of another class of IL methods named adversarial imitation learning [1, 2], which applies a totally different objective from BC.

References:

[1] Jonathan Ho and Stefano Ermon. Generative adversarial imitation learning. Advances in neural information processing systems, 29, 2016.

[2] Ilya Kostrikov, Kumar Krishna Agrawal, Debidatta Dwibedi, Sergey Levine, and Jonathan Tompson. Discriminator-actor-critic: Addressing sample inefﬁciency and reward bias in adversarial imitation learning. In International Conference on Learning Representations, 2018.

**Questions:**

1. For Battle zone and Q*bert in Figure 1.(a), for a fixed FLOP budget, how the data size change when increasing the model size?

---

> ### Author Response · Authors · 2023-11-16
> **Response to reviewer u8oj**
>
> We thank the reviewer for their valuable feedback.
>
> > Given existing works studying the scaling law in LLMs, the novelty of investigating the scaling law of BC in games is limited. As claimed in this paper, LLMs use the same MLE objective as BC. Thus, the only difference between this paper and existing works is the agent domain. As a result, it is not very surprising that the single-agent games domain can exhibit a similar scaling law. A more interesting direction is to investigate the scaling law of another class of IL methods named adversarial imitation learning [1, 2], which applies a totally different objective from BC.
>
> First, while the loss is indeed the same as for LLMs, and so perhaps it’s plausible to expect scaling laws here, **this work is the first to explicitly show that this is in fact true for imitation learning in single-agent games!**
>
> Second, the more surprising part of our paper is that **we not only get scaling laws in the loss (just like the LLMs), but also in the raw returns of the environments!** This is highly nontrivial, given return isn’t necessarily expected to scale smoothly as mentioned by Hilton et al. [3]. **Compare this with Kaplan et al. [4] and Hoffmann et al. [5] who only consider loss scaling** (see also the first few sentences of section 4.2 in the paper). In addition, we get very strong correlations between the compute-optimal loss and the corresponding return in the environment.
>
> Third, note that we also have RL results in section 4.3, which do optimize a completely different objective.
>
> Finally, **we agree investigating AIL-type methods could be super interesting as well**, but we think this probably deserves its own paper and hence we leave this for future work.
>
> > For Battle zone and Q*bert in Figure 1.(a), for a fixed FLOP budget, how the data size change when increasing the model size?
>
> Great question! When keeping the FLOP budget fixed, the data size *decreases* as the model size *increases*. Please refer to section E.1 of the appendix to see how we computed the FLOPs in Atari. Based on the FLOP equations there, one can see that if the model size increases, the FLOPs per forward-backward pass will increase as well, and hence the data size will need to decrease to keep the FLOPs constant.
>
>
> **We hope the above helps clarify any confusions the reviewer had about the paper, and potentially allows them to raise their score.**

---

> > ### Comment · Reviewer_U8oj · 2023-11-22
> >
> > Thank the authors for their detailed explanation. After reading the reviews from other reviewers and the responses from the authors, I'd like to keep my current score unchanged.

---

> > > ### Author Response · Authors · 2023-11-22
> > > **Thank you**
> > >
> > > We would like to thank the reviewer again for their feedback as well as their active engagement in the rebuttal!

---

### Official Review · Reviewer_oGwh · 2023-11-01

**Soundness:** 4 excellent
**Presentation:** 4 excellent
**Contribution:** 4 excellent
**Rating:** 8
**Confidence:** 3

**Summary:**

This paper systematically examines the scaling laws for the BC method on Atari and NetHack, exploring scaling across model size and dataset size.

**Strengths:**

- The paper is exceptionally well-written, introducing a novel study.
- It presents scaling trends on various metrics such as dev loss, returns, and final test returns.
- Scaled+BC demonstrates substantial improvement over existing methods in NetHack. This indicates that BC remains a robust baseline when applied with a suitable model and sample size.

**Weaknesses:**

While there are no significant weaknesses identified, including discussion points from the questions section (below) could enhance the robustness of the submission.

**Questions:**

1. Do the authors believe that similar scaling trends will persist in IID environments like Procgen, where training and testing distributions differ?
2. How do other offline RL baselines from NetHack (Table 2) perform when trained with the same 115B samples data?

---

> ### Author Response · Authors · 2023-11-16
> **Response to reviewer oGwh**
>
> We thank the reviewer for their valuable feedback, and are happy to hear they feel very positive about it!
>
> > Do the authors believe that similar scaling trends will persist in IID environments like Procgen, where training and testing distributions differ?
>
> **Yes! In fact, our NetHack results strongly support this claim.** This is because NetHack is procedurally generated (just like Procgen!) and incredibly stochastic (unlike Procgen, where the transition function is deterministic!), meaning multiple runs of the game will often differ substantially.
>
> > How do other offline RL baselines from NetHack (Table 2) perform when trained with the same 115B samples data?
>
> We did not run this experiment, as (1) it requires substantial resources and (2) offline RL is not the main focus of our paper. We suspect significant improvements might be possible there as well, but **leave a thorough investigation of scaling offline RL to future work.**
>
> Please let us know if you have any other questions or concerns, we'd be happy to address them!

---

> > ### Comment · Reviewer_oGwh · 2023-11-20
> > **Response from reviewer**
> >
> > Thank you for addressing my concerns. I believe this paper offers valuable insights for achieving effective performance within large-scale offline RL. In light of this, I feel it's appropriate to maintain my initial evaluation score.

---

> ### Author Response · Authors · 2023-11-20
> **Thank you**
>
> We are happy to hear the reviewer's positive sentiment towards our work. In addition, we would like to thank the reviewer again for their feedback as well as their active engagement in the rebuttal!

---

### Official Review · Reviewer_rcLB · 2023-11-02

**Soundness:** 2 fair
**Presentation:** 3 good
**Contribution:** 2 fair
**Rating:** 3
**Confidence:** 4

**Summary:**

This paper studies scaling laws in the setting of imitation learning. That is, it searches for relationships that are linear on a log-log plot of the following quantities: compute (FLOPS) & #parameters & amount of data vs. cross entropy loss & reward.

Whilst such laws are widely studied in supervised learning in various domains, this is the first work in the imitation learning setting, which is the major novelty of the paper.

The paper argues that such scaling laws do exist in several environments -- several Atari games and the more complex Nethack environment. They follow two established fitting procedures to define coefficients. Data for all experiments is generated by a single expert policy (PPO agents for Atari, and a scripted agent for Nethack). Actions are discrete, and optimized via cross-entropy.

The paper uses their laws to predict / extrapolate the ingredients for a model that should recover expert performance in Nethack, though the resulting agent falls below that level.

They offer a brief experiment using online RL (rather than imitation learning), and present scaling laws also discovered in that regime on reward.

**Strengths:**

- The objective of the paper -- studying scaling laws in BC -- is timely and I'm sure would capture the interest of many researchers in and around imitation learning.
- The paper presents a suite of experiments, and subsequent analysis, that required substantial effort to set up, and significant compute to run. I see value in sharing this with other researchers.
- Presentation is mostly clear. The main results, particularly Figure 1, come out very clean.
- The paper offers something of a counter-opinion to the recent work of Hilton et al..

**Weaknesses:**

I see several red flags in the paper -- both in the framing of the paper and the experimental design. These are severe enough that I wouldn't recommend acceptance at this time.

Major -- framing
- A casual reader could be led by the framing of the paper to believe that scaling laws hold generally across imitation learning. However, a more critical read leaves one with the impression the authors set out deliberately to uncover environments with relationships that could be classified as scaling laws in imitation learning. At its core that is the main issue with the paper -- rather than being a scientific investigation of these properties, it is a biased search for results that fit the scaling-law template.
- There are fundamental reasons to expect that such laws regarding rewards would _not_ hold generally across imitation learning -- as was the focus of Hilton et al.. Indeed there is the straightforward issue of saturation at the expert's return level, clearly seen in Figure 2's Atari curves, which invalidates any scaling laws beyond a point. There is only brief mention of this in the limitation section.

Major -- experiment design
- The authors mention that environments that do not fit the story of the paper are excluded -- Double dunk proved too simple an environment -- and games that were selected were picked due to intuitions that their score would lead to scaling laws. This drastically reduces any conclusions that can be drawn from the paper -- it shows only there exist _some_ environments where scaling laws hold for _some_ portion of the compute space.
- As I understood, learning rates were held fixed, both across and within model sizes. This has been shown to be a critically important detail in other works -- Hoffmann et al., Hilton et al.. Justification for this requires more than a small note in the limitations section.
- The true test of scaling laws is their ability to predict properties of models orders of magnitude larger than what coefficients have been fitted for. I'm pleased the authors were brave enough to take on such a test and share results. Unfortunately, the forecasted Nethack model delivered a return of 4x smaller than it was forecast, discrediting the accuracy of these scaling laws.


Other issues
- The RL portion of the paper is distracting and tangential to the main message of the paper.
- The model sizes in Atari are very small (1k to 5M params) -- it feels a little incongruous given work around scaling laws is motivated to help understand very large models.
- Model scaling was done by changing the LSTM and CNN width only. Is it not important to proportionally scale depth?
- Data was generated by a single expert algorithmic policy per environment. Whilst this is a common set up for RL benchmarks, it is far more restrictive than work on scaling laws in other domains, such as language, which are fitting data from a broad distribution of demonstrators.

**Questions:**

- See major issues within weaknesses.
- One thought that occurred to me -- scaling laws require a decomposition of reducible and irreducible error. Usually it is intractable to estimate the irreducible error in domains like language, but it would be possible in imitation learning with a fixed policy. Might this add depth to the analysis?

---

> ### Author Response · Authors · 2023-11-14
> **Response to reviewer rcLB (Part #1)**
>
> We thank the reviewer for their thorough and valuable feedback.
>
> > A casual reader could be led by the framing of the paper to believe that scaling laws hold generally across imitation learning.
>
> We would like to point out that the title of our work includes the suffix “in single-agent games”, which we deliberately put there to make it very clear **our findings are restricted to single-agent games, not imitation learning in general**. We repeat this constraint throughout the abstract and the introduction.
>
> However, within the single-agent game environments, we also specifically focus on ones with somewhat *dense* rewards, as we would like to have access to some metric of progress that we can plot when scaling up compute (i.e. this would be hard to do in sparse reward games). **To help readers better understand this limitation of our work, we have now added this assumption in several places in the beginning of the paper** (see text marked in red in the updated pdf).
>
> > However, a more critical read leaves one with the impression the authors set out deliberately to uncover environments with relationships that could be classified as scaling laws in imitation learning. At its core that is the main issue with the paper -- rather than being a scientific investigation of these properties, it is a biased search for results that fit the scaling-law template.
>
> We’re sorry to hear this is the impression the reviewer got when reading our paper. **We hope our responses below to the concrete issues the reviewer brought up can help convince the reviewer that this was certainly *not* the case.**
>
> > There are fundamental reasons to expect that such laws regarding rewards would not hold generally across imitation learning -- as was the focus of Hilton et al..
>
> We never make the claim that such laws hold generally across imitation learning, so we completely agree! **Please see the first paragraph of Section 6 (Limitations)** where we argue that our findings are likely limited to natural performance metrics.
>
> However, the limitation above doesn’t mean our settings aren’t useful! **There are plenty of interesting problems in RL which match these settings (i.e. have dense rewards).** Real-world examples of these settings include inventory management [1] and data center cooling [2]. In both problems, we can expect the reward to be pretty dense (see equation 2 in Madeka et al. [1] and equation 3 in Lazic et al. [2]).
>
> > Indeed there is the straightforward issue of saturation at the expert's return level, clearly seen in Figure 2's Atari curves, which invalidates any scaling laws beyond a point. There is only a brief mention of this in the limitation section.
>
> This is true of many scaling laws: if you extrapolate the work of Kaplan et al. [1], it will predict zero log-loss, and then keep on going to predicting negative log loss. It is clear this can’t be true since the scaling law is bounded below by the entropy of natural language. **Similarly, saturation at the expert’s return is *expected* for scaling laws in imitation learning once the required compute budget is reached.** This is because performance is bounded above by the performance of the data generating policy.
>
> **We added the clarification above to an updated version of the pdf as well.**
>
> > The authors mention that environments that do not fit the story of the paper are excluded -- Double dunk proved too simple an environment -- …
>
> We would like to clarify that **we did *not* exclude any environment because it didn’t fit the story of the paper.** On the contrary, **except for Double Dunk, we have included *every* environment we have tested so far in the paper!**
>
> Let us clarify a bit more why we decided to exclude Double Dunk from the paper. When training our family of model sizes with imitation learning on the Double Dunk dataset, we found that even our smallest model could perfectly learn the underlying policy (i.e. top-1 accuracy on validation > 99%). We would like to study settings where scaling data and model size is *meaningful*. However, if the smallest model in our family can already perfectly recover the underlying policy with relatively few samples, then there simply isn’t much to study when scaling up since recovering the underlying policy is the best you can hope for with behavioral cloning. Hence, we don’t consider this a very meaningful environment to include for our purposes.
>
> Nevertheless, **we are happy to include the plots for Double Dunk in the appendix** if the reviewer thinks this would improve the paper’s clarity and transparency about the exclusion of the Double Dunk results!

---

> ### Author Response · Authors · 2023-11-14
> **Response to reviewer rcLB (Part #2)**
>
> > ... and games that were selected were picked due to intuitions that their score would lead to scaling laws. This drastically reduces any conclusions that can be drawn from the paper -- it shows only there exist some environments where scaling laws hold for some portion of the compute space.
>
> **We did not pick environments based on intuitions their score would lead to scaling laws.** In fact, we’re not sure how one would go about this without actually running the analysis and checking.
>
> However, **our work does require assumptions on the underlying environments.** Specifically, we only test environments where the reward is at least somewhat dense, meaning we can expect learning progress to be reflected to some extent in the reward function. **This, however, is by no means a guarantee we will observe scaling laws in these environments when using imitation learning!** The reason we focus on environments with dense rewards is that theoretically, one could always construct an environment with very sparse rewards where the agent wouldn’t receive any reward until it has perfectly captured the underlying policy, at which point we would see a sudden peak. We acknowledge this sparse setting would break our current analysis, and could potentially be fixed by introducing proxy metrics different from the environment reward, as done in Hilton et al. [3] for example. However, deeply investigating proxy metrics and their corresponding scaling trends is not a focus of our work.
>
> **We have updated our pdf to include clarifications on our assumptions and on the way we picked environments** (see text in red). We hope this clarifies things for the reviewer, as well as for future readers of our work.
>
> > As I understood, learning rates were held fixed, both across and within model sizes. This has been shown to be a critically important detail in other works -- Hoffmann et al., Hilton et al.. Justification for this requires more than a small note in the limitations section.
>
> Broadly, we agree with the reviewer that more careful tuning of the learning rates would have resulted in less uncertainty about the exact numbers we find in the paper. **However, the key point here is that we do not believe this affects the major *trends* found in our work.** The reason for this is that past work has also simply relied on heuristics and empirical findings which may or may not hold in our case. Specifically:
>
> 1. Regarding learning rates *across* model sizes, the work of Kaplan et al. [4] uses a rule-of-thumb equation to scale their LR to bigger models, while Hilton et al. [3] sometimes tune their LRs per model size and sometimes rely on heuristics that scale model size with respect to initialization scale. For simplicity and to limit computational cost, our work does not consider the role of LR on scale and simply uses a constant of 1e-4 that seems to work pretty well for the model sizes and FLOP budgets we ran (i.e. we didn’t encounter any divergence issues).
>
> 2. Regarding learning rates *within* model sizes, we assume the reviewer here is referring to varying the compute budget for a fixed model size. Hoffmann et al. [5] show that the optimal thing to do here is to choose learning rate schedules per compute budget. However, we follow the work of Hilton et al. [3] and instead choose to use “snapshots” due to computational expense. As pointed out in the limitations section of our paper, this does mean there is uncertainty in the exact values of our power law exponents, as is the case for Hilton et al. [3].
>
> > The RL portion of the paper is distracting and tangential to the main message of the paper.
>
> This section already mostly lives in the appendix, with only the core result in the main paper. While removing it is an option, we’d prefer not to because of the following reasons.
>
> 1. This section provides a nice preliminary extension of our results to RL. This is particularly exciting since analyzing RL can be more difficult than the supervised setting.
>
> 2. It connects our work to that of Hilton et al. [3] and shows that even in complex environments like NetHack the reward function can act as a natural performance metric.
>
> 3. None of the other reviewers seemed to mind this addition, and in fact reviewer fMKL seemed to appreciate these results.
>
> **If later in the rebuttal it turns out more reviewers share this concern, we’re happy to make some adjustments here.** However, so far, it seems like this section is interesting enough to justify a quarter page of our paper.

---

> ### Author Response · Authors · 2023-11-14
> **Response to reviewer rcLB (Part #3)**
>
> > The true test of scaling laws is their ability to predict properties of models orders of magnitude larger than what coefficients have been fitted for. I'm pleased the authors were brave enough to take on such a test and share results. Unfortunately, the forecasted Nethack model delivered a return of 4x smaller than it was forecast, discrediting the accuracy of these scaling laws.
>
> While we agree our results would have been stronger if we had achieved the predicted return of 10k exactly, **we do *not* believe this error discredits the underlying scaling law *trends***. If we think back to the work of Kaplan et al. [4], we realize they made several errors as well, such as the following:
>
> - They find slope coefficients which are inconsistent with the later work of Hoffmann et al. [5]
>
> - They used suboptimal heuristics for determining hyperparameters
>
> - They find scaling width vs. depth doesn’t matter as much, though recent work finds jointly scaling width and depth is better than scaling either individually (see Figure E.3 in Wortsman et al. [6])
>
> Given these flaws in the original work of Kaplan et al. [4], do we still find this paper useful for the scientific community? We very much believe so! While their work didn’t capture all aspects of scaling laws correctly, it still made the community aware of stark scaling law trends in neural language models. **Similarly, our work may have various places where errors accumulate which causes the 10k vs. 2.7k error (e.g. suboptimal hyperparameters, uncertainty in the predicted power law coefficients, etc.), however we don’t believe this discredits the underlying scaling trends we find in the paper.**
>
> In addition to having provided the perspective above, **we would also like to point the reviewer to Appendix J where we performed a rolling time series cross-validation of all power law plots in the paper**. If we look at the “BC maximal returns vs. FLOPs” entry in Table 7, we can see that the RMSE is only 390, much lower than the 10k - 2.7k difference for our forecasting experiments. Hence, we argue that, for the FLOP budget we test in Figure 2a, our scaling laws are accurate predictors of future performance. Furthermore, we also plot the confidence intervals of all regression parameters when performing cross-validation in Figures 15 - 18 of Appendix J. If we look at the last confidence intervals of each plot, we can see that the slope coefficients never include 0, meaning they are all statistically significant.
>
> > The model sizes in Atari are very small (1k to 5M params) -- it feels a little incongruous given work around scaling laws is motivated to help understand very large models.
>
> The goal of our work is to investigate the role of scale in imitation learning for single-agent games. To that end, we would argue there should be no prior preference over a particular range of model sizes: the model sizes should be dictated by what makes sense for the environments we’re studying. **In the case of Atari, Figure 1a (Battle Zone and Q*bert) tells us that running model sizes larger than 5M parameters is not very meaningful because they will be further and further away (towards the right) from the minimum of the parabola.** Note, however, that this is different for NetHack, and hence we do run larger model sizes there.
> Also, note that model sizes in sequential decision making settings tend to be a lot smaller than for language (which is where most of the scaling laws work happens).
>
> > Model scaling was done by changing the LSTM and CNN width only. Is it not important to proportionally scale depth?
>
> This is an interesting point, and seems to **still be an active area of research!** Kaplan et al. found very minor sensitivities to scaling depth vs. width, while the more recent work of Wortsman et al. [6] indicates it’s important to *jointly* scale depth and width (see Figure E.3). Hilton et al. [3] perform mostly width-scaling though they do some depth-scaling as well for Procgen (but they never try *joint* scaling), however they do not find any approach to be conclusively better than the other. It’s possible our results could be improved by more careful scaling of width vs. depth, but **we leave a thorough investigation of this to future work.**
>
> > Data was generated by a single expert algorithmic policy per environment. Whilst this is a common set up for RL benchmarks, it is far more restrictive than work on scaling laws in other domains, such as language, which are fitting data from a broad distribution of demonstrators.
>
> We agree this is an interesting setting to study! **We would be happy to run our analysis on a new dataset** for one of our Atari games where samples are collected by multiple expert PPO policies (with different seeds), or potentially even by experts with different underlying learning algorithms (PPO, DQN, etc.). Would the reviewer find this insightful?

---

> > ### Author Response · Authors · 2023-11-14
> > **Response to reviewer rcLB (Part #4)**
> >
> > > One thought that occurred to me -- scaling laws require a decomposition of reducible and irreducible error. Usually it is intractable to estimate the irreducible error in domains like language, but it would be possible in imitation learning with a fixed policy. Might this add depth to the analysis?
> >
> > While we agree with the reviewer it would be possible to estimate the irreducible loss term by evaluating the data-generating policy on the validation set, we’re not sure if this would add much depth to the analysis. **However, if the reviewer could be more concrete about what insight this would add, we would be happy to reconsider adding it!**

---

> ### Comment · Reviewer_rcLB · 2023-11-21
> **Rebuttal response**
>
> Thank you for your response, I appreciate the effort you have put into it. I am inclined to agree with you in several places around the work being valuable, even if the execution is imperfect, for example in response to using a fixed learning rate, and the inaccurate coefficients in NetHack.
>
> My main reservation still persists -- that the paper seeks to fit power laws to relationships that do not appear (to me) to fit this template well. This is particularly true for the relationship with reward. I like the added references to `dense rewards' in several places in the paper, but I think this is too vague -- it seems to be a central requirement for power laws to hold, which is the topic of the paper. What is a precise definition of a dense reward? Dense in what respect? What environments and reward functions obey this dense assumption? What happens to the relationships when this assumption is stretched and broken?
>
> I don't have further questions at this time. I plan to continue the discussion with the other reviewers.

---

> > ### Author Response · Authors · 2023-11-21
> > **Response to Reviewer rcLB**
> >
> > We would like to thank the reviewer for their active engagement in the rebuttal!
> >
> > > My main reservation still persists -- that the paper seeks to fit power laws to relationships that do not appear (to me) to fit this template well. This is particularly true for the relationship with reward.
> >
> > We would like to reiterate that, for all our environments, the power law *does* describe these relationships pretty well, for both loss and reward, as can be seen from figures 1 & 2 (and more in Appendix I). See the limitations regarding reward in Section 6.
> >
> > > I like the added references to `dense rewards' in several places in the paper, but I think this is too vague -- it seems to be a central requirement for power laws to hold, which is the topic of the paper. What is a precise definition of a dense reward? Dense in what respect? What environments and reward functions obey this dense assumption? What happens to the relationships when this assumption is stretched and broken?
> >
> > These are wonderful questions, but we argue that every one of these could encompass a full paper. As it stands, our paper is already pretty dense, and so trying to include these questions into the current work wouldn't do them justice. **However, we do believe it's worth for future work to dive into these questions, and we hope our work can inspire the community to do so!**

---

### Author Response · Authors · 2023-11-14
**Global list of references**

Below we provide a list of references we use for all responses to all reviewers.

**References:**

1. Madeka, Dhruv, et al. "Deep inventory management." arXiv preprint arXiv:2210.03137 (2022).

2. Lazic, Nevena et al. “Data center cooling using model-predictive control.” Neural Information Processing Systems (2018).

3. Hilton, Jacob, Jie Tang, and John Schulman. "Scaling laws for single-agent reinforcement learning." arXiv preprint arXiv:2301.13442 (2023).

4. Kaplan, Jared, et al. "Scaling laws for neural language models." arXiv preprint arXiv:2001.08361 (2020).

5. Hoffmann, Jordan, et al. "Training compute-optimal large language models." arXiv preprint arXiv:2203.15556 (2022).

6. Wortsman, Mitchell, et al. "Small-scale proxies for large-scale Transformer training instabilities." arXiv preprint arXiv:2309.14322 (2023).

7. Rajaraman, Nived et al. “On the Value of Interaction and Function Approximation in Imitation Learning.” Neural Information Processing Systems (2021).

8. Chen, Jianyu et al. “Deep Imitation Learning for Autonomous Driving in Generic Urban Scenarios with Enhanced Safety.” 2019 IEEE/RSJ International Conference on Intelligent Robots and Systems (IROS) (2019): 2884-2890.

9. Kim, Geon-Hyeong, et al. "Demodice: Offline imitation learning with supplementary imperfect demonstrations." International Conference on Learning Representations. 2021.

---

### Author Response · Authors · 2023-11-21
**Gentle reminder for discussion.**

Dear reviewers,

Thank you again for your valuable and constructive feedback. They were insightful and have allowed us to improve the paper (see updated paper draft with a different color for each reviewer). **Since the discussion period is closing soon, we encourage all reviewers who haven't replied to our latest responses to check those out and provide their response.** We would greatly appreciate it. Thank you to those reviewers who have already replied.

Please also let us know if there is any additional clarifications or experiments we can provide to further show the merit of our paper!

Thank you.

---

### Meta-Review · Area_Chair_d1QY · 2023-12-07

**Metareview:**

This paper explores the applicability of power laws to behavior cloning in single-player games. The study is extensive but currently critically flawed by a potential confirmation bias. Instead of excluding results that do not fit the narrative, it would significantly improve the contribution to study why the scaling laws do not apply in these cases. The core hypothesis and topic of the paper sparked significant interest with reviewers suggesting this further work would be valued widely by the research community if the authors choose to pursue it further in future work.

**Justification For Why Not Higher Score:**

+ Critical issues in the methodology (particularly results excluded for not fitting the narrative)

**Justification For Why Not Lower Score:**

N/A

---

### Decision · Program_Chairs · 2024-01-16

Reject